# Vitamin C Regulates the Profibrotic Activity of Fibroblasts in In Vitro Replica Settings of Myocardial Infarction

**DOI:** 10.3390/ijms24098379

**Published:** 2023-05-06

**Authors:** Yichen Xu, Huabo Zheng, Pakhwan Nilcham, Octavian Bucur, Felix Vogt, Ioana Slabu, Elisa Anamaria Liehn, Mihaela Rusu

**Affiliations:** 1Department of Intensive Care Medicine, University Hospital, RWTH Aachen, Pauwelsstr. 30, 52074 Aachen, Germany; 2Department of Cardiology, Angiology and Intensive Care, University Hospital, RWTH Aachen, 52074 Aachen, Germanypnilcham@ukaachen.de (P.N.);; 3Institute of Molecular Medicine, University of Southern Denmark, Campusvej 55, 5230 Odense, Denmark; 4“Victor Babes” National Institute of Pathology, Splaiul Independentei nr. 99-101, Sector 5, 050096 Bucharest, Romania; 5Viron Molecular Medicine Institute, 1 Boston Place, Ste 2600, Boston, MA 02108, USA; 6Institute of Applied Medical Engineering, Helmholtz Institute, Medical Faculty, RWTH Aachen University, 52074 Aachen, Germany; 7National Heart Center Singapore, 5 Hospital Dr., Singapore 169609, Singapore

**Keywords:** vitamin C, antioxidant capacity, fibroblast, gene transcription regulator, collagen modulator

## Abstract

Extracellular collagen remodeling is one of the central mechanisms responsible for the structural and compositional coherence of myocardium in patients undergoing myocardial infarction (MI). Activated primary cardiac fibroblasts following myocardial infarction are extensively investigated to establish anti-fibrotic therapies to improve left ventricular remodeling. To systematically assess vitamin C functions as a potential modulator involved in collagen fibrillogenesis in an in vitro model mimicking heart tissue healing after MI. Mouse primary cardiac fibroblasts were isolated from wild-type C57BL/6 mice and cultured under normal and profibrotic (hypoxic + transforming growth factor beta 1) conditions on freshly prepared coatings mimicking extracellular matrix (ECM) remodeling during healing after an MI. At 10 μg/mL, vitamin C reprogramed the respiratory mitochondrial metabolism, which is effectively associated with a more increased accumulation of intracellular reactive oxygen species (iROS) than the number of those generated by mitochondrial reactive oxygen species (mROS). The mRNA/protein expression of subtypes I, III collagen, and fibroblasts differentiations markers were upregulated over time, particularly in the presence of vitamin C. The collagen substrate potentiated the modulator role of vitamin C in reinforcing the structure of types I and III collagen synthesis by reducing collagen V expression in a timely manner, which is important in the initiation of fibrillogenesis. Altogether, our study evidenced the synergistic function of vitamin C at an optimum dose on maintaining the equilibrium functionality of radical scavenger and gene transcription, which are important in the initial phases after healing after an MI, while modulating the synthesis of de novo collagen fibrils, which is important in the final stage of tissue healing.

## 1. Introduction

An MI is an extreme heart pathological condition caused by cardiomyocytes’ exposure to a prolonged lack of oxygen, thereby, leading to their death [1]. The following remodeling processes induce structural, compositional, biomechanical changes in both uninjured and injured regions of the heart, dramatically affecting heart’s biomechanics and function [2,3,4]. Cardiac remodeling processes such as scar and fibrosis formation are seen as evidence of primary cardiac fibroblasts’ activation and differentiation that serve as the main ECM-producing cells. Their conversion into structural and functional matrix – components is crucial to rapidly compensate for tissular loss and heart’s function [2,5]. During all these processes, the ECM undergoes dramatical changes from provisional matrix containing predominately fibronectin [6] to initial abnormal accumulations of types I and III collagen, followed by a continuous type I collagen robust fibrils generation for completely mature matrix formation. For example, at early time points, after an MI, the provisional non-collagen matrix is enriched with adhesive fibronectin components [7] to avoid tissular discontinuity and possible heart rupture, while large amounts of structurally robust collagen are synthesized to prevent heart failure at later time points of heart healing [8,9]. Therefore, a therapeutic strategy to influence cardiac fibrosis needs to be initiated at an early time point before activation of myofibroblasts [10] and should be continued during the performance of myofibroblasts’ activity at a later time point after an MI [11,12,13]. 

While the current treatment implies angioplasty with reperfusion of the ischemic areas, it seems that the sudden oxygen level increase can be toxic for cells, increasing ROS formation, oxidative stress, and cellular damage [14,15]. There are no data that show that reperfusion could have an effect on scar formation and ECM synthesis compared with chronical ischemia [16,17]. Type I collagen is one of the major contributors to a mature scar formation, and it is known that vitamin C may influences its synthesis mainly via its co-enzymatic function [18,19]. Complementarily, during healing events after an MI, the intracellular antioxidant capacity decreases under exacerbated oxidative stress and it may reside in various threshold levels of ROS, which downstream the regulation of wound healing [20]. It is known that ischemia/reperfusion is a supplementary source of increased ROS which causes progressive cell death and tissular degradation [14,15].

Vitamin C, initially identified as the central radical scavenger role player, became very popular for its pleiotropic applications in recent clinical studies in cardiovascular diseases [21,22,23] or cancers [24]. However, in the MI context, vitamin C seemed to have an unelucidated protective role on cells survival, not only as ROS scavenger, but also as an essential regulator in collagen synthesis in cardiac fibroblasts. Moreover, recent clinical trials have shown that vitamin C is important in providing the maintenance of α-tocopherol [23], as well as the reduced state of riboflavin [25]. Moreover, vitamin C also showed the potential of improving the oxidative imbalance and vascular remodeling caused by SARS-CoV-2 infections, achieving the relief of symptoms such as fatigue among patients with long-term COVID by reducing the endothelial barrier’s permeability [26]. However, the association of low vitamin C levels with an increased risk of a coronary heart diseases is still controversial [27,28], and many studies using vitamin C supplementation have delivered inconsistent results [29,30,31]. Despite the fact that many preclinical studies highlighted the potential of vitamin C to accelerate tissue healing by increasing the synthesis of type I collagen [32], most of the clinical studies target neutrophils [33] and oxidative stress [34], while having less focus on the effects of vitamin C on the structural remodeling phases of heart such as: inflammation, proliferation, and maturation. 

Therefore, we plan here to identify the detailed mechanisms and role of vitamin C in healing processes at the cellular level in conditions mimicking healing after an MI. The purpose of this study is to establish the effect of vitamin C on primary cardiac fibroblasts’ functionality in response to collagen synthesis. 

## 2. Results

### 2.1. Assessment of Effect of Vitamin C Activity on Primary Cardiac Fibroblast Metabolism and Treatment Optimization

To assess the effect of the specific activity of vitamin C on primary cardiac fibroblasts’ function, we initially established the onset in vitro conditions of vitamin C. The establishment of optimum onset in vitro conditions is essential when assessing the bioenergetic status of mitochondria. In normoxia and hypoxia settings, the rate of O_2_ consumption is controlled by the activity of the electron transport chain and adenosine triphosphate (ATP) production, processes coupled via the H^+^ gradient and leakage (Figure 1 and Appendix A). In order to investigate the specific contribution of vitamin C on ATP synthesis and O_2_ consumption states in both normoxia and hypoxia settings, we coupled oligomycin (an inhibitor of ATP synthesis) and carbonyl cyanide-p-trifluoromethoxyphenylhydrazone (FCCP) (a dissipater of H^+^ gradient) with various concentrations of vitamin C (0, 10, 30, and 50 μg/mL), which were added to the culturing medium. Under basal conditions, vitamin C enhanced the metabolic profile of mitochondria at a concentration of 10 μg/mL by two-fold more in normoxia than it did in hypoxia (Figure 1A,B). At a concentration of 10 μg/mL of vitamin C, the basal and maximal respiration levels of ROS increased approximately twice in normoxia compared to that of hypoxia (20 ± 1.88 pmoles/min/μg, **** *p* ≤ 0.0001) (Appendix A). At a similar concentration of vitamin C, ATP production was two-fold more stimulated in normoxia (13.91 ± 0.48 pmoles/min/μg, **** *p* ≤ 0.0001) than it was in hypoxia (Appendix A) whereas the H^+^ leak values were significantly reduced in hypoxia (0.35 ± 0.001 pmoles/min/μg, **** *p* ≤ 0.0001) (Appendix A) compared with those in normal conditions. Minimal variation in the spare capacity levels between normoxia (10.23 ± 0.001 pmoles/min/μg, * *p* ≤ 0.05) and hypoxia was estimated at 10 μg/mL, which was compared to those of the rest of the groups (**** *p* ≤ 0.0001) (Appendix A). At a vitamin C concentration of 10 μg/mL, nonmitochondrial respiration was not altered, although increased concentrations of vitamin C (30 μg/mL and 50 μg/mL) reduced significantly the nonmitochondrial respiration in both normoxia and hypoxia settings (**** *p* ≤ 0.0001) (Appendix A). 

A concentration of vitamin C of 10 μg/mL proved its efficiency at the mitochondrial level both in normoxia and hypoxia settings. Therefore, for the supplementation of primary cardiac fibroblasts culture media with vitamin C, a concentration of 10 μg/mL was considered to be the onset value for further experiments.

### 2.2. The Radical Scavenging Role of Vitamin C 

Since vitamin C is a water-soluble antioxidant, by scavenging numerous reactive oxidant species (ROS), we have investigated under normal vs. profibrotic conditions the effect of vitamin C on intracellular and mitochondria ROS (iROS and mROS) accumulation in primary cardiac fibroblast cell cultures on different mimic coatings. These experiments confirmed that at a concentration of 10 μg/mL, vitamin C acts as a radical scavenger. The coating composition replicates different physio-pathological conditions during healing after an MI: fibronectin (early and provisional matrixes), fibronectin–collagen (intermediary and transition matrixes), and collagen (mature scar). We defined the profibrotic settings in those conditions, which include in addition to those of hypoxia (low O_2_ (2%) supply) transforming growth factor (TGF-β1), fibronectin, fibronectin–collagen co-assembled, and collagen coatings. Thus, we timely estimated the expressions of intracellular ROS and mitochondria ROS by dihydroethidium (DHE) and mitochondrial superoxide indicators (mitoSOX) oxidation, respectively, and normalized them to DAPI (Figure 2 and Figure 3, and Appendix A). In the absence of vitamin C, iROS formation was improved in normal conditions (24 h) on fibronectin-containing coatings at a maximum of 0.23 ± 0.04%; **** *p* ≤ 0.0001 (Figure 2A and Appendix A), and it stabilized to comparable values of the control (0.23 ± 0.04%) at 72 h, independent of the coating composition. Interestingly, in the presence of vitamin C, iROS formation was equally upregulated as in the control on fibronectin-composed coatings, mostly at later time points (24 h: 0.023 ± 0.004%; 72 h: 0.1 ± 0.062%, **** *p* ≤ 0.0001) (Figure 2B and Appendix A). In contrast to the control, mROS expression was shortly upregulated at early incubation times, and it significantly decayed at later time points on each coating (Figure 2C and Appendix A), which may indicate the potential effects of ECM scaffolds on mitochondrial activity. While fibronectin suppressed mROS formation at 72 h after early upregulation, collagen-containing coatings significantly improved mROS formation over time (Figure 2D and Appendix A).

While only the profibrotic environment moderately influenced the ROS formation, vitamin C in profibrotic settings drastically downregulated ROS formation (Figure 3A–D and Appendix A). In the profibrotic environment, in contrast to the control, the rate of iROS expression significantly increased in primary cardiac fibroblasts at later time points, independent of the type of coating (Figure 3A and Appendix A). Vitamin C supplementation downregulated iROS expression at later time points (from a maximum of 0.196 ± 0.026 at 24 h to a minimum of 0.073 ± 0.008 **** *p* ≤ 0.0001) (Figure 3B and Appendix A). fibronectin–collagen Vitamin C showed its steady antioxidant capacity over time only on the fibronectin–collagen coating (Figure 3B and Appendix A). Similarly, while mROS expression showed a decay trend over time in profibrotic conditions (Figure 3C and Appendix A), particularly on the collagen coating, the cell culture augmented with vitamin C resulted in the additional downregulated expression of mROS on each coating type (Figure 3D and Appendix A). 

Altogether, vitamin C under normal conditions was able to reduce mROS levels at the later time points only on the fibronectin coating. Under profibrotic settings, vitamin C reduced significantly the amount of both iROS and mROS with time—a powerful radical scavenging process that appeared to be independent of the coating composition. These findings suggest that the antioxidant effect of vitamin C might be potentiated in healing stages in a specific manner after an MI. In order to avoid different perceptions, the OY scale of all the graphs in Figure 2 and Figure 3 was specified at 0.6.

### 2.3. Substrate Composition Modulate Fibrillar Collagen Synthesis in Normal Settings

Given the tight correlation between fibrotic tissue formation and the upregulated state of collagen in activated primary cardiac fibroblasts, we analyzed in depth the effect of profibrotic settings on primary cardiac fibroblast differentiation towards the myofibroblast phenotype (SM22) regarding the synthesis of fibrillar types I, III, and V collagen. Primary cardiac fibroblasts’ synthesis activity was assessed at 24 h and 72 h of incubation on fibronectin, fibronectin–collagen, and collagen by estimating double-stained cells with DAPI-SM22, as well as DAPI-type I procollagen/type I/III/V collagen (Appendix A). In normal settings and in the absence of coatings, primary cardiac fibroblasts abundantly expressed SM22 (Figure 4A and Appendix A). Surprisingly, SM22 upregulation was not detected on any other substrate type (Figure 4A and Appendix A). The primary cardiac fibroblasts’ activity on various coating types was reflected in a steady decreasing tendency of col1a1 gene expression over time on collagen-containing coatings, whereas in the control group and on fibronectin, it remained constant (Figure 4B). Further translational pathways in regard to collagen synthesis revealed the association expression between type I procollagen (Figure 4C and Appendix A) and type I collagen (Figure 4D and Appendix A) on collagen substrates, indicating its potentiating role in collagen synthesis. Although the amount of type I procollagen was moderately estimated on the remaining coatings (Figure 4C and Appendix A), the expression of type I collagen increased over time, independent of the coating types (Figure 4D and Appendix A). A prominent variation at the late time point in which Col3a1 mRNA expression occurred was detected on fibronectin–collagen mix coatings (Figure 4E), which reflected the significant upregulation of type III collagen expression (0.671 ± 0.013%, **** *p* ≤ 0.0001) (Figure 4F and Appendix A). Surprisingly, col5a1 mRNA expression followed a similar behavior to that of col3a1 mRNA on each coating type (Figure 4G). Although at the protein level, the level of type V collagen expression was two orders of magnitude lower than those of other collagen types, on collagen coatings at 72 h, it was about two-fold more upregulated than that at 24 h (0.057 ± 0.006%, **** *p* ≤ 0.0001) (Figure 4H and Appendix A). Taking together, these results represent the overview translation of mature collagen fibers formation from cellular to gene and protein levels (Figure 4I).

### 2.4. Active Primary Cardiac Fibroblasts Turned into Profibrotic-Driven col5a1gene Expression

Although similar ranges of expression values are found in both normal and profibrotic settings, SM22’s expression behavior under profibrotic conditions significantly differs from that in normal conditions (Figure 5A). Following a moderate increase over time on each coating, SM22 reached a maximum expression over time on fibronectin coatings at 72 h (0.02 ± 0.002%; **** *p* ≤ 0.0001), demonstrating that fibronectin is required at every incubation time for primary cardiac fibroblasts to remain activated (Figure 5A and Appendix A). On the contrary, on the fibronectin–collagen mix, primary cardiac fibroblast activity did not vary with time (Figure 5A and Appendix A), but primary cardiac fibroblasts became activated on the rest of the coatings. The expression levels of SM22 and col1a1 mRNA (Figure 5B) remained constant in time only on the fibronectin–collagen mix (0.55 ± 0.08%, **** *p* ≤ 0.0001). The fibronectin-collagen mix was the only substrate that showed its steady suppressing potential effect on SM22 and col1a1 mRNA expressions, which may indicate the intimate interactions between substrate types -cell culture conditions and collagen formation. Regardless of the translation of mRNA levels under profibrotic settings into proteins, type I procollagen was evaluated via immunohistochemistry (Figure 5C and Appendix A). The secretion levels of type I procollagen decreased one order of magnitude more under profibrotic settings than those estimated in normal conditions did (Figure 5C and Appendix A). This behavior might potentially explain the rapid conversion of type I collagen (Figure 5D). Independent of the coating composition, the profibrotic settings suppressed the col3a1 mRNA expression values to one order of magnitude lower than those in normal conditions (Figure 5E). However, fibronectin and non-collagen-based coatings potentiated the expression type III collagen at the later time point (0.5 ± 0.02%, **** *p* ≤ 0.0001) (Figure 5F and Appendix A). It is noteworthy that while the amplification of col5a1 mRNA expression remained constant over time on each coating, it increased about four-fold more under profibrotic conditions vs. control group and normal conditions (4.44 ± 0.37% vs. 0.71 ± 0.08%) (Figure 5G). High levels of col5a1 mRNA, in turn, seemed to be required for protein translation, though they were of two orders of magnitude smaller than those of other collagen types. The protein expression levels of type V collagen increased under profibrotic settings only in the presence of collagen and the fibronectin–collagen mix (Figure 5H and Appendix A), indicating that these coatings assist type V collagen synthesis. 

Taken together, these data evidence the potentiating role of fibronectin-based coatings in primary cardiac fibroblasts activation, and notably, type III collagen synthesis. On the contrary, collagen-based coatings seemed to be especially required during types I and V collagen secretion. These findings are summarized in Figure 5I. 

### 2.5. Vitamin C Regulates Timely the Turnover of De Novo Fibrillar Collagen in Normal Settings

In similar settings, we investigated the effect of vitamin C on the synthesis of de novo fibrillar subtypes of collagen, which originate from activated primary cardiac fibroblasts. In normal conditions, primary cardiac fibroblasts exhibited constant SM22 expression levels (0.073 ± 0.005%) (Figure 6A and Appendix A) and displayed a similar trend to those in vitamin C - free treatment (0.03 ± 0.001%) (Figure 6A and Appendix A). While col1a1 mRNA expression on fibronectin remained constant over time, on collagen-based coatings, it significantly increased over time (Figure 6B). Furthermore, the converted fraction of type I collagen from its precursor is separately potentiated by vitamin C by 1.3-fold and 1.7-fold only at the later time point on the collagen coating (Figure 6C,D and Appendix A). Interestingly, independent of the coating strategy, barely any expression variation was estimated for type I collagen (Figure 6D) compared to that in the vitamin C-free settings (Figure 4D). This might suggest that vitamin C neither potentiates the translational rates from genes levels nor has a potential structural and/or compositional compensating role in type I collagen. Contrary to the vitamin C-free system (Figure 4E), the supplementation of cells’ medium with vitamin C seemed to preserve the col3a1 mRNA expression levels over time (Figure 6E), whereas the protein level of type III collagen (Figure 6F, Appendix A) is comparable with that of vitamin C-free systems (Figure 4E,F). Similarly, vitamin C constantly maintained col5a1 mRNA expression levels over time (Figure 6G), while the expression levels of type V collagen (Figure 6H and Appendix A) followed a similar trend to that of the vitamin C-free treatment. These aspects regarding stimulating effect of vitamin C on fibrils’ assembly are summarized in Figure 6I. 

### 2.6. Vitamin C Effect Associated De Novo Fibrillar Collagen Synthesis Depends on Coating Composition 

Under profibrotic conditions, vitamin C induced the rapid and significant activation of primary cardiac fibroblasts, mostly on fibronectin and collagen coatings (Figure 7A and Appendix A). Col1a1 mRNA expression was significantly upregulated by vitamin C only on the fibronectin–collagen mix coating (2.027 ± 0.33%, **** *p* ≤ 0.0001) (Figure 7B), while the conversion of type I procollagen increased by approximately 19- or 23-fold on fibronectin coatings, which was compared to those caused by the vitamin C-free treatment (Figure 7C and Appendix A). Particularly on collagen coatings, type I procollagen exhibited the lowest expression level (0.003 ± 0.001%, **** *p* ≤ 0.0001). Interestingly, vitamin C significantly upregulated collagen type I levels on fibronectin coatings at the later time point (0.618 ± 0.041%, ** *p* ≤ 0.01) (Figure 7D and Appendix A), which was not observed in the primary cardiac fibroblasts that did not undergo the vitamin C treatment (Figure 6D). In contrast, col3a1 mRNA expression (Figure 7E) increased in the presence of vitamin C, mostly on the fibronectin–collagen mix coatings (0.090 ± 0.25%, **** *p* ≤ 0.0001). At the protein level, the expression level of type III collagen significantly increased at the later time point on collagen-based coatings (Figure 7F and Appendix A). The effect of vitamin C on col5a1 mRNA expression was evident in the presence of fibronectin and fibronectin–collagen mix coatings, as it increased by between 7- and 8-fold more than that in the case of vitamin C-free systems (8.413 ± 0.374% on fibronectin, **** *p* ≤ 0.0001 and 7.485 ± 0.090% on the fibronectin–collagen mix, *** *p* ≤ 0.001) (Figure 7G and Figure 4G). However, the increased value of col5a1 gene expression is not associated with small values of collagen type V (0.033 ± 0.003%, * *p* ≤ 0.05) (Figure 7H and Appendix A). 

Under profibrotic settings, we identified in vitro a potential dual function of vitamin C in fibrillar collagen synthesis, such as: a stimulator of col5a1 mRNA gene transcription patterns, and at the same time, a stabilizer of col1a1 mRNA and col3a1 mRNA, particularly on collagen coatings. Furthermore, at the protein level, vitamin C seemed to structurally and compositionally stabilize type I and III collagen on the fibronectin-based coatings, while type V collagen remained constant over time independent, of the coating composition. The prominent features of primary cardiac fibroblasts’ activation and the exacerbated fibrillary build-up structure of de novo collagen in the presence of vitamin C are schematically represented in Figure 7I.

## 3. Discussion

In the present in vitro study, we revealed novel functions of vitamin C besides its (i) antioxidant capacity, such as: (ii) sustaining the mRNA transcription of ECM proteins, (iii) it is a structural and compositional modulator, as well as temporal regulator, of the synthesis of de novo fibrillar subtypes of collagen, which originate in activated primary cardiac fibroblasts under profibrotic settings. The temporal and compositional settings mimicking heart healing conditions included TGF-β1-dependent cardiac fibroblasts’ activation and differentiation and three types of 2D scaffolds such as: fibronectin—mimicking the initial provisional matrix of early healing phase; collagen–fibronectin mix—corresponding to an intermediary matrix of early-to-late healing phase; collagen—associated with mature scar of late healing phases. Moreover, in the MI context, vitamin C seemed to have an unelucidated protective role on cell survival not only as ROS scavenger [21,22,23], but also as an essential regulator in collagen synthesis in cardiac fibroblasts [32].

Our findings suggest that hypoxia impairs the mitochondrial respiration rate by reducing ATP production. At low doses, vitamin C showed its antioxidant power (radical scavenger), causing increased mitochondrial respiration. The cytotoxic effect of vitamin C at high doses on mitochondrial activity might be an effect of changing its functional balance from an antioxidant that is more prone towards a co-enzymatic regulator role in collagen synthesis. A concentration function approach of vitamin C excludes smaller concentrations (e.g., 1–2000 μM) at which vitamin C alone showed a poor cell protecting effect [35]. Moreover, at low concentrations, the antioxidant function of vitamin C seems to be primarily hindered as it is chemically unstable at the physiological pH [36].

As a result, the gene regulator function of vitamin C decreases with the increasing affinity of coatings to generate an optimum gene expression level required in the associative inflammatory phase after an MI. For example, under profibrotic conditions, vitamin C orchestrates the tail-to-tail polymerization of col3a1 and col5a1 genes: a process which seemed to be cooperatively potentiated by both fibronectin and collagen coatings. Moreover, vitamin C may sustain types I and III collagen by enabling their assembling with type V collagen, hence, evidencing its key role for the initiation of fibrillogenesis. Yet, this process has neither been evidenced in vitro nor in vivo, and its role is systematically discussed below.

### 3.1. Differentiation

Our data indicate that at the cellular level, a concentration of 10 μg/mL of vitamin C is sufficient to activate and differentiate primary cardiac fibroblasts, independent of the coating composition and normal or profibrotic settings. It is known TGFβ1 induces the transdifferentiation of fibroblasts to myofibroblasts and contribute to the progression of fibrosis [37]. Our results pointed out the role of vitamin C to also promote primary cardiac fibroblasts’ phenotype switching in myofibroblast phenotype and collagen synthesis [38]; however, there are no data to show any correlation between the vitamin C and TGFβ1. In our study, we first evidenced that vitamin C promotes the activation and differentiation of isolated primary cardiac fibroblasts under both normal and profibrotic settings. Though, the activation and differentiation of primary cardiac fibroblasts were particularly enhanced on fibronectin and, to a moderate extent, on collagen coatings under profibrotic conditions. This suggests that the signaling pathway mediated by fibronectin from external transduction [7] appears to be amplified by the presence of vitamin C. Thereby, we may expect an increased expression of collagen fibrils on adhesive fibronectin coatings as a consequence of the increased interaction between fibronectin and, e.g., type III collagen [39]. 

### 3.2. Antioxidant 

Under particular profibrotic settings, vitamin C protected primary cardiac fibroblasts against ROS formation. In normal conditions, on both fibronectin- and collagen-rich matrixes, the antioxidant mechanism of the cell itself seemed to vanish over time, thereby increasing the accumulation of more mROS than iROS under profibrotic settings over time. A good detoxification balance of both cellular compartments is essential and required for optimal cell function. Our data agree to the existent data that indicate that the dose of vitamin C can have opposite, anti- and prooxidative effects on cells. A lower dose of Vitamin C (5.3 μg/mL) acts as an antioxidant by increasing the activity of Mn-SOD and GPx in mitochondria [40]. Two hundred micrometers (35.2 μg/mL) of vitamin C can decrease the ROS level [41]. In thyroid cancer cells, 1 mM (176 μg/mL) vitamin C showed toxic effects, being reported to inhibit the activity of catalase and SOD in ARO [42,43], or it can lead to formation of a large amount of ROS and cause the cell cycle arrest of human bladder carcinoma T24 cells [44] and Ehrlich carcinoma cells [45,46,47]. In the HT-1080 fibrosarcoma cell line, while 0.5 mM (88 μg/mL) decreased the rate of activity, 1.2 mM (211 μg/mL) vitamin C greatly induced the ferroptosis of HT-1080 fibrosarcoma cell line [48,49]. Rouleau et al. also shows that 5–20 mM of vitamin C (880–3520 μg/mL) has a highly toxic effect on Hepatocellular carcinoma cells (Hep G2 cell line) in a H_2_O_2_-dependant manner [50]. 

In our mimicking systems, vitamin C proved its scavenging activity to remove ROS from both cellular compartments in a timely manner to achieve optimum cell function. The scavenging mechanism of ROS seemed to be independent of the coatings’ composition. The antioxidant activity of vitamin C over time in isolated primary cardiac fibroblasts was, for the first time, evidenced in mimicking healing conditions after an MI. These findings may suggest the long-term protective role of vitamin C by preventing damages that occur at the cell and organ levels. From ROS data, one can suggest that vitamin C may lead to adaptive reprogramming mitochondrial metabolism and physiological energy balance regulation, which are necessary for cells’ surviving and fibrillar collagen synthesis. However, vitamin C is not stable under normal conditions and normal oxygen concentration in the environment [51,52,53]. Therefore, vitamin C can lose total or partly its function to scavenge ROS in normal conditions.

### 3.3. Sustaining mRNA Transcription of ECM Protein 

Hypoxic conditions lead to 70% of different subtypes of collagen transcriptional changes, such as fibrillar collagen mRNAs—col1a1, col3a1, and col5a1 [54]. Thus, abnormal modifications of collagen mRNA expression result in significant myocardial function threats [55]. 

In normal conditions, vitamin C seemed to sustain col3a1 and col5a1 mRNA gene transcription and their conversion into associated proteins. The constant expression of col3a1 and col5a1 mRNA over time was, for the first time, revealed, and it may indicate the fact that vitamin C is an essential regulator of gene transcription. The sustaining of collagen mRNA gene transcription seemed to be synergized at early time points. The overlapping functions of vitamin C as a free radical scavenger and an mRNA gene transcription regulator might preferentially be exercised in the early heart healing phase to protect and restore highly dynamic cells’ activity associated with collagen synthesis. 

Surprisingly, under profibrotic settings, vitamin C sustains the level of *col1a1 mRNA* on fibronectin and *col5a1 mRNA* and *col3a1 mRNA* on fibronectin and collagen coatings. However, much to our surprise, *col5a1 mRNA* is significantly upregulated over time, independent of the coating composition. This finding reveals the role of vitamin C under pro-fibrotic settings in modulating collagen synthesis by increasing *col5a1 mRNA* expression levels, as type V collagen is known to actively initiate fibrillogenesis. This is a characteristic that is in line with previous studies related to TGF-β1 and vitamin C’s effect on increased transcription of *col1a1* genes [56,57]. In comparison to normal settings, on fibronectin–collagen co-assembles under pro-fibrotic conditions, vitamin C potentiated the expression of *col1a1 mRNA* in a timely manner. 

This finding might ascertain the ischemic heart regions needed for compensating for impaired EF (%) by tuning the composition and accumulation of de novo type I collagen fibers [2]. Moreover, the decrease in *col3a1 mRNA* expression and the significantly increased expression of *col5a1 mRNA* under profibrotic conditions might indicate the more significant re-organization of *col3a1* and *col5a1* genes than that which occurs in the normal conditions. Similar findings were observed in vitro, indicating the tail-to-tail organization of *col3a1* and *col5a2* genes [58]. This specific gene organization might further be modulated at 72 h by vitamin C on fibronectin-based coatings, as presumably, fibronectin and collagen cooperatively potentiate the polymerization of collagen fibrils [59]. The polymerization process might further be assisted by scavenging intra- and extracellularly ROS products to protect primary cardiac fibroblasts activation from, e.g., necrosis. As a result, the gene regulator function of vitamin C decreases with an increasing affinity of coatings to generate the optimum gene expression level required in the associative inflammatory phase after an MI.

### 3.4. Modulator of Fibrillar Collagen Synthesis 

The cardiac scar and fibrosis development is a process of abnormal ECM remodeling seen in cardiac remodeling after MI. The remodeling processes consist of a series of timed molecular and cellular events of which primary cardiac fibroblasts’ activation and differentiation have a central role to rapidly compensate tissular loss and the heart’s function by synthesizing timely ECM components [2,5]. For example, at early time points, after an MI, the provisional non-collagen matrix is enriched with adhesive fibronectin components [7] to avoid tissular discontinuity and the heart’s possible rupture, while large amounts of structurally robust collagen are synthesized to prevent heart failure at later time points during the heart’s healing [8,9]. Therefore, a therapeutic strategy to influence cardiac fibrosis is necessary to be initiated at an early time point before activation of myofibroblasts [10], and should be continued during myofibroblasts’ activity at a later time point after an MI [11,12,13]. Vitamin C has an essential role in healing processes by decreasing ROS accumulation and in fibrosis formation by regulating collagen synthesis in an associative manner [60,61].

We evidenced that under normal settings, the coating composition-dependent feedback mechanism seemed to keep the non-proportional conversion degree of type I procollagen into type I collagen high. It is noteworthy to mention that if the vitamin C-free conversion synthesis is highly influenced by the coating composition, vitamin C regulates this conversion at the later time point by reducing the expression of type I procollagen to its equilibrium conversion, independent of the coating composition. The equilibrium conversion of type I collagen was timely accompanied by the upregulation of type III collagen and the steady synthesis of type V collagen, though in smaller amounts. Notably, on fibronectin-based coatings, type III collagen is highly upregulated by vitamin C, whereas type V collagen is significantly upregulated on collagen coatings. In addition, the reciprocal molecular interactions between collagen and collagen–fibronectin regulate de novo fibrils formation, a process that counts also in collagen fibrillogenesis, and thereby in fibrotic tissue formation [62]. Hence, it can be hypothesized that vitamin C and/or the combined coating composition and vitamin C are essential for fibrillar collagen formation, presumably by facilitating the nucleation of minor amounts of type V collagen in type III collagen, which is a process that is required for optimal fibrillary formation and high-quality heart tissue. However, how the combined coating composition and vitamin C control the procollagen conversion into stable, cross-linked collagenous protofibrils needs to be further investigated. 

The profibrotic conditions significantly stimulated the primary cardiac fibroblast’s function. The increased secretion of collagen in response to type I procollagen expression might indicate the conversion repression mechanism of type I procollagen into type I collagen, a process controlled by vitamin C. Yet, this conversion has been neither evidenced in vitro nor in vivo, and thereby, it requires in-depth investigations. Though the expression of type I procollagen might be suppressed by, e.g., side products produced due to a poor catalyzing post-translational hydroxylation reaction or by defect self-assembly of collagen molecules [63]. The undesired hydroxylated precursors of collagen might intracellularly accumulate, which may lead to unstable triple helical structures. In this hypothesis, other than the abundance accumulation and disturbance of the synthesis/degradation equilibrium of collagen, the dysregulation of the triple helical collagen structure might be critical in fibrotic tissue formation. Vitamin C sustains the activation of primary cardiac fibroblasts by removing ROS formation and dynamically regulates de novo type III collagen formation, particularly on fibronectin, which is associated with early inflammatory-to-proliferative phase after an MI. In the later mimicking reparative phase, vitamin C demonstrates its stabilizing role for an appropriate balance between types I and III collagen fibrils, leading, thereby, to a mature scar formation.

However, the decrease in the rate of type V collagen synthesis to one order of magnitude more than those of types I and III collagen may indicate its ability to reinforce the structural composition of these fibrillar collagen, a process that is apparently independent of the coating composition. These findings are in line with studies revealing the interplay of profibrotic conditions and fibronectin on collagen fibrillogenesis [38,64]. In murine mice, the role of type V collagen was speculated for the first time in vivo and discussed in terms of the key regulator factor for collagen fibrillogenesis [2]. Under oxidative stress conditions, vitamin C seemed to have a central modulator role in collagen conversion from their precursors, which is a process that might also involve the ability of vitamin C to be an active enzyme complement for collagen synthesis, structure, and composition stabilization. The importance of vitamin C in stabilizing the collagen gene expression map has also emerged by dynamically modulating the secretion of fibrillar types I, III, and V collagen, which has not been emphasized under normal settings nor in the absence of vitamin C. Although the therapeutic targets of vitamin C and its underlying mechanisms in cardiovascular diseases remained to be elucidated, our studies reveal the pleiotropic activity of vitamin C for cell protection and optimum collagen fibril formation and maturation – critical remodeling processes during inflammation-to-proliferation-to maturation phases after MI. 

### 3.5. Advantages

Compared to the multitude of targeted antifibrotic therapies being investigated [65,66], the safety profile of vitamin C has a major advantage. In vivo, the activity of vitamin C might switch on/off or these states might complement each other at various healing time points after an MI, thereby demonstrating its considerable tissue specificity. For example, it can be hypothesized that a lot of the ability of vitamin C to scavenge free radicals and to regulate the mRNA expression during inflammation is presumably complemented by its structural regulating role in collagen and regulating function of post-translational hydroxylation processes during proliferative-to reparatory remodeling phase, as demonstrated in the present study. Moreover, the in vitro/in vivo supplementation of vitamin C may lead to the adaptive reprogramming of the mitochondrial metabolism and physiological energy balance regulation, which are necessary for cell survival and fibrillar collagen synthesis.

### 3.6. Study Limitation

This is a preliminary study that was conducted for assessing vitamin C’s role in regulating remodeling in a culture model of healing conditions after an MI, which aimed to create the context for further in vivo experiments. It seems that the complex pharmacological method used to deliver vitamin C to mice to modulate remodeling after an infarction requires preliminary in vitro assays to select and predict accurately potential in vivo adverse effects, as high doses of vitamin C showed potentially cytotoxic effects on primary cardiac fibroblasts activation. A previous study on vitamin C and riboflavin in guinea pigs has shown that the peroxidation of dehydroascorbic acid with H_2_O_2_ produces a new organic peroxide, which leads to apoptosis and alveolar rupture [25]. However, the cellular model reality is biased because it is still challenging to simulate in vitro the cytokines and chemokines action during immune processes that undergo real-time changes in vivo. Moreover, most aspects of the temporal expressions of mRNA and proteins might give rise to various proportions of interferent side products such as mRNA fragments, which make difficult to accurately reflect collagen synthesis. In addition, the effective concentration bias of vitamin C due to its fast metabolization or degradation [22] needs further consideration, as its in vitro functions may not reflect entirely the metabolic demands of the body.

## 4. Materials and Methods

### 4.1. Preparation of Coatings for Cell Seeding

The coatings for cell seeding were chosen to mimic the pathophysiological environment during healing after an MI. Fibronectin is the predominant coating during the early phases, constituting the provisional matrix. The fibronectin-rich matrix undergoes a transformation into mixed content of fibronectin and collagen until a final, mature scar matrix containing predominantly collagen is formed. Fibronectin (human plasma; Sigma-Aldrich; San Jose, CA, USA), self-assembled fibronectin–collagen, and collagen G (Sigma-Aldrich; San Jose, CA, USA) were used as scaffolds to mimic ECM remodeling undergoing inflammation, proliferation, and maturation post-MI. stock solutions of 4 mg/mL and 1 mg/mL for collagen and fibronectin were prepared in Dulbecco’s phosphate-buffered saline (DPBS; Sigma-Aldrich; San Jose, CA, USA) and mixed for 5 min to homogeneity in order to obtain final working concentrations of 100 μg/mL and 10 μg/mL, respectively, according to the manufacturer’s protocol. Collagen was previously mixed to homogeneity to a final pH of 3.6. Co-assembles of fibronectin–collagen were obtained by adding equal volumes from each working solution to coat 24-well plates. Coatings were annealed for 1h in incubator at 37 °C to accelerate the crosslinking process. The excess compound was gently removed by aspiration and carefully rinsed twice with DPBS before seeding the cells.

### 4.2. Cell Isolation, Culture, and Treatments 

Primary cardiac fibroblasts were isolated from wild-type C57BL/6 mice by mechanical trituration and enzymatic digestion (Collagenase type Ⅰ; Roche; Basel, Switzerland) using well-established protocols, as described before [67]. Isolation procedures were performed according to the European legislation and approved by local German authorities (AZ: 8.87-50.10.35.09.088). The primary cardiac fibroblasts pellet obtained by centrifugation (350× *g*, 20 °C, 5 min) was incubated for 1 h in normoxia (21% O_2_, 5% CO_2_, 37 °C, Innova@ CO-48 Incubator, New Brunswick Scientific, Enfield, CT, USA). By excluding cardiac cells, endothelial cells or macrophage/mesenchymal cells, all of which requires special culturing media and factors, and by removing floating cells, we considered the remaining viable cells of primary cardiac fibroblast phenotype in our culture model. Moreover, we preferred to use cardiac fibroblasts and not just normal fibroblasts, as those are available to humans, since there are many recent studies indicating the significant differences between cardiac fibroblasts and other types of fibroblasts [68,69,70]. Adherent fibroblasts were further cultured in Dulbecco′s modified eagle′s medium: high glucose (DMEM, Sigma-Aldrich; San Jose, CA, USA) containing 1% Penicillin/Streptomycin (PAA–The Cell Culture Company, Cölbe, Germany) and 10% serum (Fetal bovine serum dialyzed; PAN Biotech, Aidenbach, Germany). On each coating type, primary cardiac fibroblasts were seeded at a density of 20,000 cells/well. After 24 h of incubation, primary cardiac fibroblasts underwent normoxia. In parallel, the primary cardiac fibroblasts culture was supplemented with 15 μg/mL transforming growth factor β stimulation (TGF-β1; Cell Signaling Technology, Danvers, MA, USA) to activate differentiation towards myofibroblast and incubated in hypoxia (2% O_2_) in a designated incubator (2% O_2_, 5% CO_2_, and 93% N2 at 37 °C, Innova@ CO-48 Incubator, New Brunswick Scientific, Enfield, CT, USA) for 24 h. Fresh vitamin C (l-ascorbic acid 2-phosphate sesquimagnesium hydrate; Sigma-Aldrich, Darmstadt, Germany) solution of 10 mg/mL was prepared in PBS and added at a working concentration of 10 μg/mL to the primary cardiac fibroblasts exposed to hypoxia or normoxia and further incubated for 24 and 72 h, respectively. Studies showed that the primary cardiac fibroblasts took 72 h to change their phenotype and ECM proteins [2]. Thus, we conducted all analysis in vitro at 24 h to evaluate potential changes in gene and mRNA expression, as well as at 72 h to confirm any modification in protein and ECM protein expression, but also to examine whether the downregulation of genes and mRNA expression is transitory under vitamin C stimulation.

### 4.3. Analysis of Mitochondrial Activity 

To determine the appropriate concentration of vitamin C for an optimum functionality of mitochondria, 20,000 cells/well cardiac fibroblasts were seeded in a 96-well seahorse plate and incubated in the absence and presence of vitamin C at various concentrations: 0 μg/mL, 10 μg/mL, 30 μg/mL, and 50 μg/mL. The magnitudes of key parameter changes in mitochondrial respiration (Basal respiration, maximal respiration, ATP production, Proton leak, Spare capacity, and non-mitochondrial respiration) were monitored by means of Seahorse XFe96 Analyzer, Agilent. After reaching confluence, primary cardiac fibroblasts were incubated in normoxia/hypoxia without/with 0.2 mL (15 μg/mL) TGF-β1 to activate their differentiation towards myofibroblast. After 24 h incubation time, cells were supplemented with vitamin C at various concentrations and for 1 hour in both normoxia and hypoxia. The supernatant was replaced by 180 μL DMEM supplemented with 25 mM glucose, 2 mM glutamine, and 1 mM pyruvate (pH 7.4). A standard mitochondrial stress test was performed using 1 µM oligomycin, 3 µM FCCP, and 0.5 µM rotenone + 0.5 µM antimycin A. The procedure was carried out according to the standard protocol provided by the manufacturer: 3 measurements per phase, acute injection followed by 3 min mixing, 0 min waiting, and 3 min measuring. The total protein was isolated by 0.1% Trition-X100 and quantified by using the DC Protein Assay kit (BioRad, München, Germany) using an Infinite M200 microplate reader (TECAN, Männedorf, Switzerland). The maximum absorbance was read at 700 nm. Data were exported from Seahorse Wave 2.2.0 and further analyzed using the Prism software (Version 8.02).

### 4.4. Immunofluorescence 

Primary cardiac fibroblasts cultured on fibronectin, fibronectin–collagen mix, and collagen coatings were fixed with 4% paraformaldehyde for 10 mins (PFA, Sigma-Aldrich; San Jose, CA, USA). After blocking with 1% bovine serum albumin (BSA, biomol, Hamburg, Germany) and horse serum in PBS for 1 h, cells were incubated overnight with diluted primary antibodies: SM22 (ab14106, Abcam, Cambridge, UK) as marker of myofibroblast differentiation [71], type I collagen (CL50151AP-1, Cedarlane, Burlington, ON, Canada), type III collagen (ab7778, Abcam, Cambridge, UK), type V collagen (ab7046, Abcam, Cambridge, UK), type I procollagen (sc-8782, Santa Cruz Biotechnology, Dallas, TX, USA) at 4 °C, followed by the corresponding secondary antibodies and counterstained with 4′,6-Diamidino-2-phenylindole (DAPI, H-1200, Vector, Berlin, Germany). Image overlay and densitometric analysis were performed using ImageJ software (National Institute of Mental Health, Bethesda, MD, USA). The amount of each type of collagen, including type I procollagen, was normalized to the total number of cells/images. This method was preferred due to the insufficient protein amount resulted from isolations procedure for eventual Western blot analysis. 

### 4.5. RNA Isolation and Gene Expression

RT-qPCR was employed to assess (24 h and 72 h) the relative expression levels of mRNA of types I, III, and V collagen in normoxia and hypoxia conditions in a timely manner in the presence and absence of vitamin C. Total RNA was isolated from primary cardiac fibroblasts using TRIzol (15596026, Life Technologies; Carlsbad, CA, USA) as indicated by the manufacturer. cDNA was synthesized from total RNA using a cDNA reverse Transcription Kit (K1691, Life Technologies; Carlsbad, CA, USA) according to the manufacture’s protocol. RT-qPCR was performed using 500 ng of cDNA per 20 μL reaction with FastStart Master (4710452001, Sigma-Aldrich; San Jose, CA, USA) and conducted on ViiA 7 Real-Time PCR System (AB Applied Biosystems, Foster City, CA, USA). The subsequent PCR conditions consisted of 40 cycles of denaturation at 95 °C for 15 s and annealing at 60 °C for 1 min per cycle. Primers were designed using Primer Blast (NCBI’s primer design tool), and information on the primers used for each gene is listed in Table 1. The specificity of primers was verified with the NCBI primer blast tool and analyzed by producing a melting curve to avoid primer dimers and non-specific amplification. The total mRNA levels are expressed as thousands of mRNA copies/μg total RNA. The expression of each gene was calculated by the 2^−ΔΔCT^ method [72], with the target gene normalized to reference β-actin gene. Each experiment was performed in triplicate and repeated 3 times.

### 4.6. Assessment of Mitochondrial and Intracellular ROS 

Dihydroethidium (D7008, DHE, Sigma-Aldrich; CA, USA) and a Red Mitochondrial Superoxide Indicator (M36008, MitoSOX, ThermoFisher Scientific, Langerwehe, Germany) were used to detect intracellular superoxide ions (O_2_^•−^) (iROS) and mitochondrial O_2_^•−^ (mROS), respectively. Treated primary cardiac fibroblasts incubated in normoxia and hypoxia and supplemented or not with 10 ng/mL vitamin C were incubated for 15 min in PBS containing 0.5 mL of 10 µM DHE or 0.5 mL of 5 µM MitoSOX, respectively, according to the manufacturer’s protocol. After washing, primary cardiac fibroblasts were fixed for 10 min with 4% PFA, followed by DAPI staining. Images were acquired using DISKUS (Hilgers, Königswinter, Germany) and subsequently analyzed via image overlay and densitometry using the ImageJ software. The amounts of iROS and mROS were normalized to the total number of cells/images.

### 4.7. Data Statistics

For immunofluorescence images, ImageJ software was used to perform quantification analysis on each iROS, mROS and immunofluorescence image. ImageJ was used to calculate the total pixels, average pixel, and cell number. The average pixel intensity was calculated for all pixels within threshold regions. The average number of pixels per 300 cells was calculated by normalizing the measured integrated density (positive region pixels*300 cell) of fluorescence signals to the number of cells (a.u.). Data points of immunofluorescent images, illustrating reduced primary cardiac fibroblasts’ activity, were not considered for further analysis. As such, the reduction or amplification of detection variance among all data was avoided [73]. Statistical analysis was performed with Prism5 software (GraphPad Software, San Diego, CA, USA) using 2-way ANOVA test, followed by Newman–Keuls post hoc comparisons. Data are presented as mean ± SEM values. In order to avoid different perceptions, the OY scale of all graphs in Figure 2 was specified at 0.6. *p*-values of <0.05 were considered to be significant.

Overview of the workflow, including the inclusion criteria and methodologies is schematically represented in Figure 8.

## 5. Conclusions 

At the cellular level, our in vitro study evidenced that vitamin C modulates the quality of types I, III, and V collagen fibrils via its temporal synergistic multivalence functionality: (i) the radical scavenger function of both mROS and iROS fractions is highly potentiated over time at the level of the oxidative protection of primary cardiac fibroblasts activation, particularly under profibrotic settings; (ii) it is an essential regulator and stabilizer of collagen mRNA gene expression in primary cardiac fibroblasts, particularly on a fibronectin–collagen coating; at the later time point, significant amounts of col5a1 mRNA translate type V collagen in small fractions, (small fractions of type V collagen can be an indicative of self-assembly with other collagenous structures); (iii) it is a highly qualitative heterostructure and composition modulator of de novo types I and III collagen fibril formation on a fibronectin–collagen coating (Figure 9). Here, vitamin C may sustain types I and III collagen by enabling their interactions with type V collagen, which is responsible for the initiation of fibrillogenesis. Focusing only on single key factor, such as an antioxidant, it may fail to fully capture the complex relevant role of vitamin C in clinical practice. Collectively, due to the ample amount of evidence of the temporal complementary functionalities of vitamin C in vitro, antioxidant–collagen synthetic regulator interplay needs to be further investigated for an optimal dose and timing as a potential preconditioner and combined therapeutic tool in cardiovascular diseases. 

## Figures and Tables

**Figure 1 ijms-24-08379-f001:**
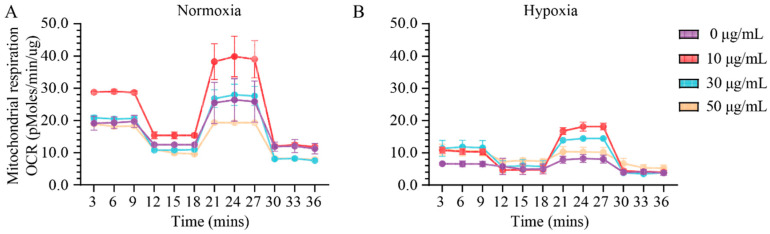
Odd and even effects of low and high vitamin C concentrations stimulate or disrupt the bioenergetic status in primary cardiac fibroblasts. A concentration of 10 μg/mL of vitamin C selectively increases the respiration profile of mitochondria, whereas at higher concentrations, the nonmitochondrial respiration is significantly reduced in normoxia (**A**) and in hypoxic conditions (**B**). Data are normalized to total protein. Error bars indicate standard error of the mean; n = 3 independent experiments per condition.

**Figure 2 ijms-24-08379-f002:**
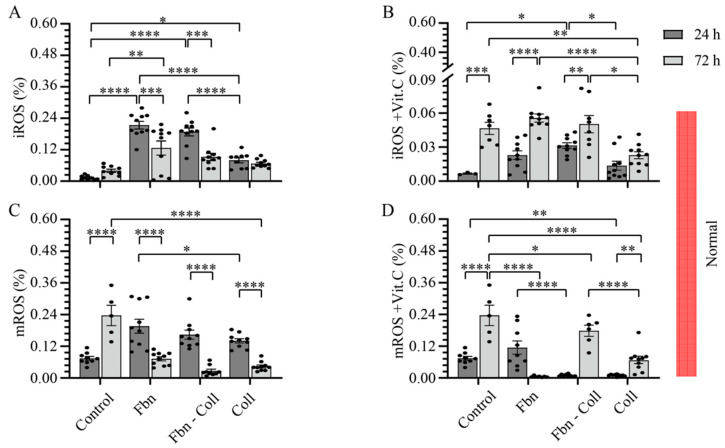
Fibronectin potentiates the radical scavenging role of vitamin C under normal settings. iROS formation is shortly upregulated on fibronectin-containing coatings (**A**). Vitamin C significantly potentiates the formation of iROS at the late incubation time, independent of coating composition (**B**). Similarly, mROS formation is shortly upregulated, independent of coating composition (**C**). Vitamin C scavenges timely mROS only on fibronectin coating, but not in primary cardiac fibroblasts cultured on collagen-containing coatings (**D**). Both iROS and mROS were detected via DHE–mitoSOX–DAPI co-staining of primary cardiac fibroblasts (10,000 cells/well). Error bars indicate standard error of the mean; * *p* ≤ 0.05; ** *p* ≤ 0.01; *** *p* ≤ 0.001; **** *p* ≤ 0.0001. n = 3 experiments per condition. Fbn (Fibronectin); Coll (Collagen); mROS (mitochondrial reactive oxygen species); iROS (intracellular reactive oxygen species).

**Figure 3 ijms-24-08379-f003:**
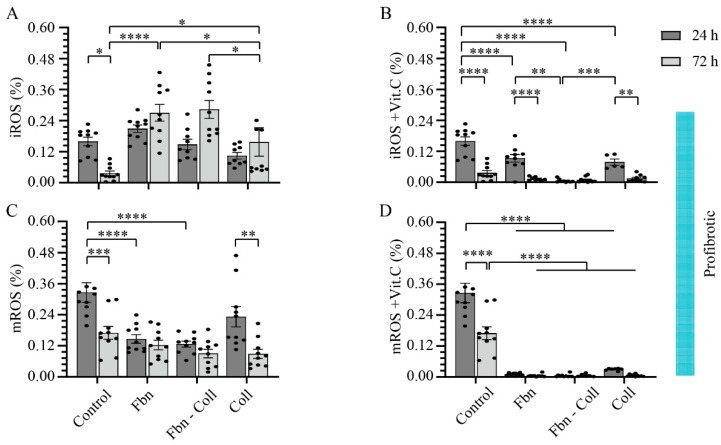
Profibrotic conditions potentiate the radical scavenging functionality of vitamin C. iROS formation is upregulated in a timely manner (**A**). Vitamin C downregulates iROS formation at the late incubation time (**B**). Similarly, mROS follows a decay trend in time (**C**), and it vanishes even more in the presence of vitamin C (**D**). Likewise, iROS and mROS were detected via DHE–mitoSOX–DAPI co-staining of primary cardiac fibroblasts (10,000 cells/well). Error bars indicate standard error of the mean; * *p* ≤ 0.05; ** *p* ≤ 0.01; *** *p* ≤ 0.001; **** *p* ≤ 0.0001. n = 3 experiments per condition. Fbn (Fibronectin); Coll (Collagen); mROS (mitochondrial reactive oxygen species); iROS (intracellular reactive oxygen species).

**Figure 4 ijms-24-08379-f004:**
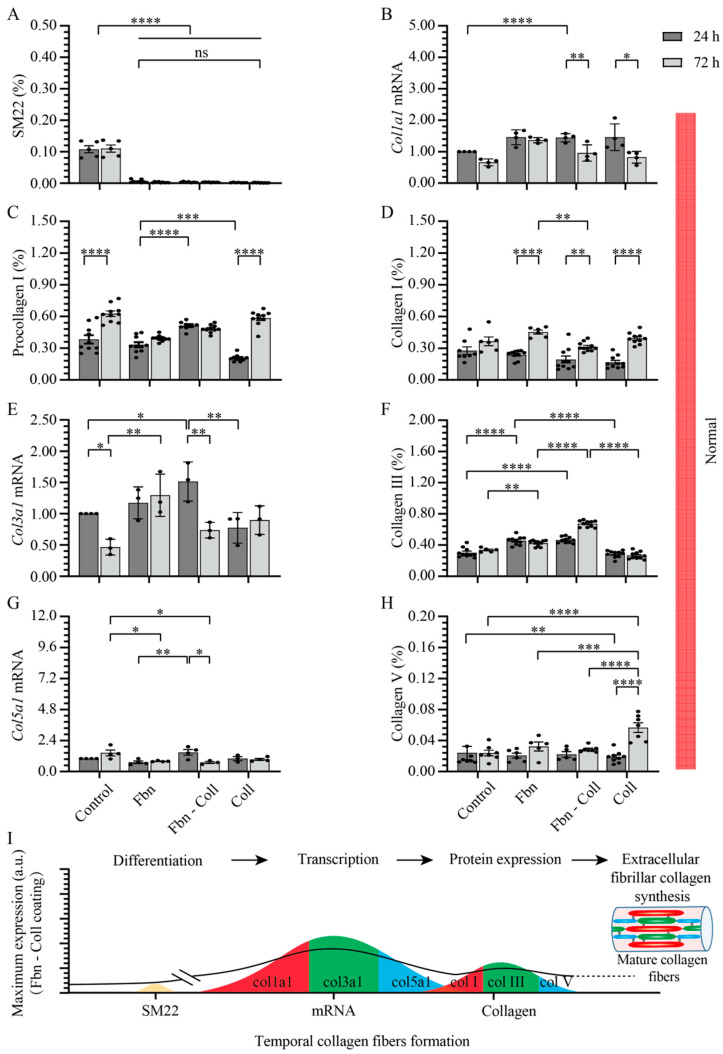
Synthesis of fibrillar collagen depends moderately on substrate composition and is not associated with primary cardiac fibroblasts’ activity. In normal settings, SM22 expression levels are significantly upregulated in control group and decrease systematically with the incubation time on different substrates (**A**). Col1a1 mRNA expression was upregulated at the early stage of incubation and decreased later only on collagen-based coatings (**B**). Type I procollagen expression remained constant on fibronectin coatings (**C**), whereas at the protein level, independent of coatings’ composition, type I collagen expression increases at 72 h (**D**). Fibronectin-based coatings significantly potentiate both col3a1 mRNA (**E**) and type III collagen expressions (**F**). Col5a1 mRNA expression is upregulated at 24 h on fibronectin–collagen mix (**G**) whereas collagen coatings promote the upregulation of type V collagen at the later incubation stage (**H**). Schematic overview of results over time. The solid line towards mature collagen fibers formation serves as a guide to the reader (**I**). Error bars indicate standard error of the mean; ns stands for not significant *p*-value, * *p* ≤ 0.05; ** *p* ≤ 0.01; *** *p* ≤ 0.001; **** *p* ≤ 0.0001. n = 3 experiments per condition. Fbn (Fibronectin); Coll (Collagen).

**Figure 5 ijms-24-08379-f005:**
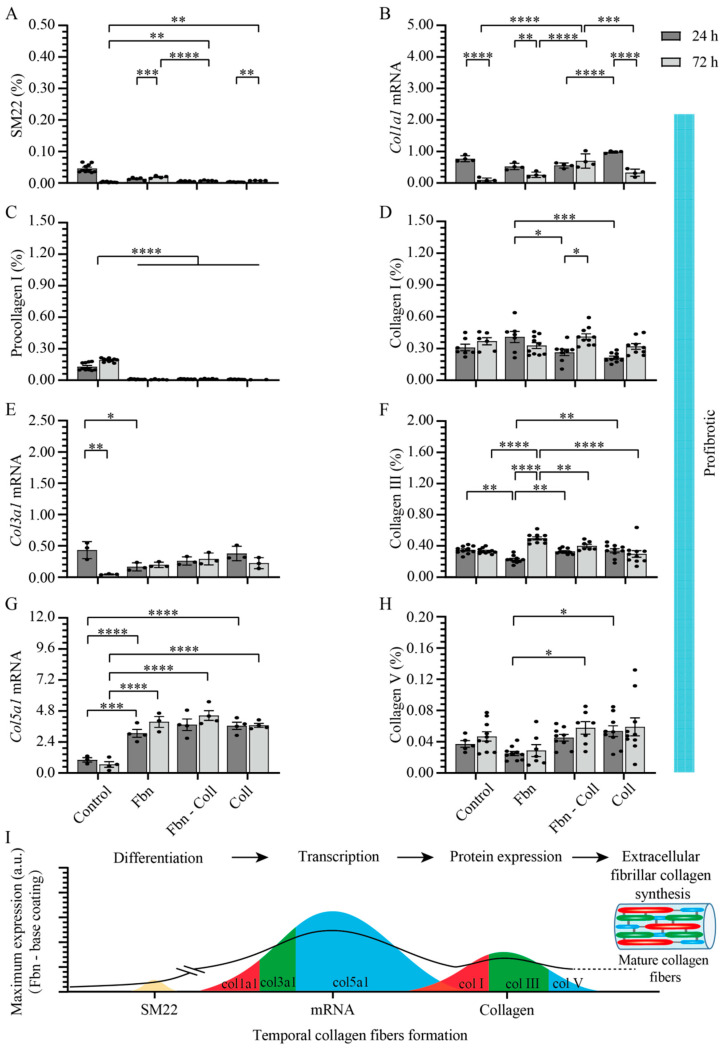
Profibrotic settings stimulate col5a1 collagen gene transcription. Profibrotic settings stimulated SM22 expression over time mostly on fibronectin coatings (**A**). The mix of fibronectin–collagen coatings barely influenced the expression of col1a1 mRNA in time, but it decreased on fibronectin and collagen coatings in a timely manner (**B**). The expression level of type I procollagen barely varied over time, but it decreased more then that of the control group (**C**). Type I collagen increased in a timely manner only on collagen-containing coatings (**D**). Col3a1 mRNA expression was suppressed in activated primary cardiac fibroblasts on each substrate (**E**), whereas fibronectin and non-collagen-based coatings potentiated the expression type III collagen at the later time point (**F**). Strikingly, col5a1 mRNA expression increased about 4-fold more than that of control group on each coating (**G**), whereas on collagen-based coatings, type V collagen protein expression stayed elevated (**H**). Schematic overview of results over time. The solid line towards mature collagen fibers formation serves as a guide to the reader (**I**). Error bars indicate standard error of the mean; * *p* ≤ 0.05; ** *p* ≤ 0.01; *** *p* ≤ 0.001; **** *p* ≤ 0.0001. n = 3 experiments per condition. Fbn (Fibronectin); Coll (Collagen).

**Figure 6 ijms-24-08379-f006:**
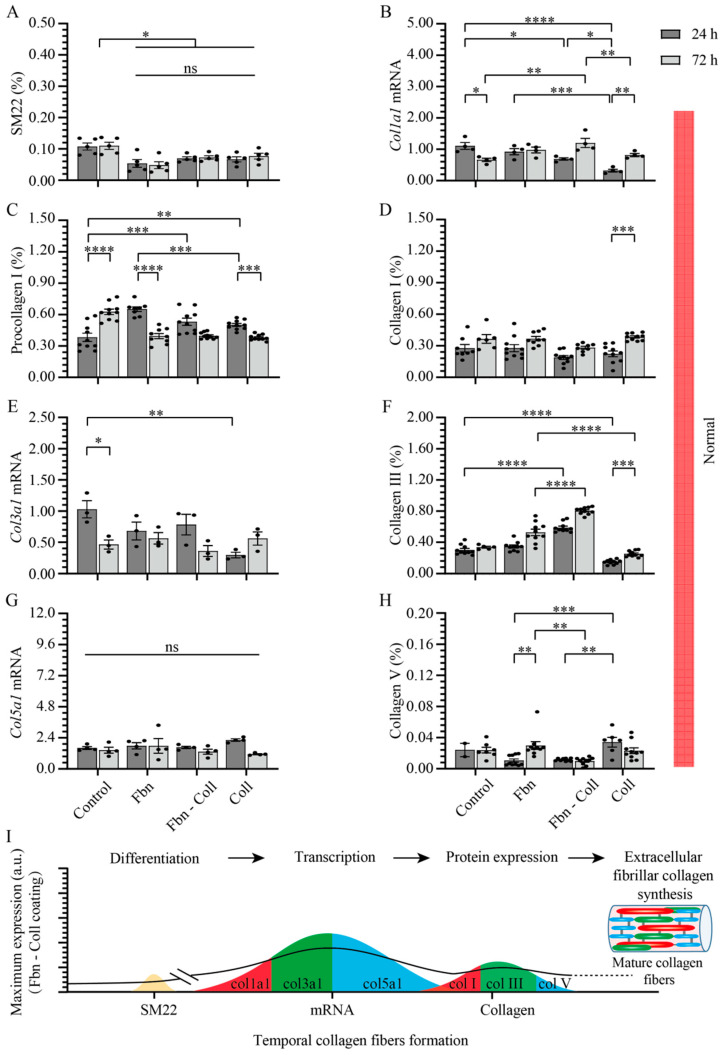
Vitamin C potentiates type III collagen turnover at the later time point under normal settings. SM22 expression levels is significantly upregulated in control group and decreases after incubation on different substrates (**A**). Col1a1 mRNA expression is enhanced, mostly on fibronectin–collagen mix and collagen coatings (**B**). Type I procollagen expression downregulates at the later time point, independent of coating composition (**C**). Type I collagen expression increases over time, peaking on fibronectin–collagen mix (**D**). The expression levels of col3a1 mRNA are constant over time, independent of coating composition (**E**), whereas the amount of type III collagen significantly increases over time (**F**). Col5a1 mRNA expression displays constant levels over time, independent of coating composition (**G**). Protein conversion levels of type V collagen reaches maximum values at the later time point on collagen coatings (**H**). Schematic overview of results, in time. The solid line towards mature collagen fibers formation serves as a guide to the reader (**I**). Error bars indicate standard error of the mean; ns stands for not significant *p*-value, * *p* ≤ 0.05; ** *p* ≤ 0.01; *** *p* ≤ 0.001; **** *p* ≤ 0.0001. n = 3 experiments per condition. Fbn (Fibronectin); Coll (Collagen).

**Figure 7 ijms-24-08379-f007:**
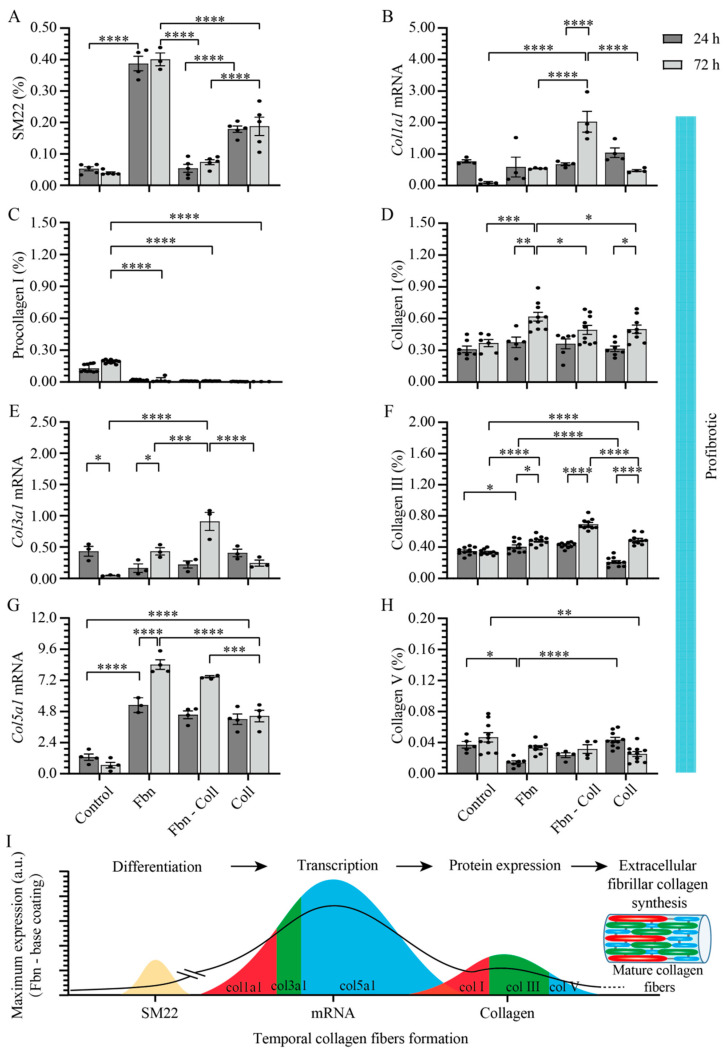
Collagen synthesis is modulated in a timely manner by vitamin C. Under profibrotic settings, vitamin C downregulates SM22 expression in primary cardiac fibroblasts cultured on collagen-based coatings (**A**). Col1a1 mRNA expression decreases in time on collagen coatings, but is significantly upregulated at the later time point on the fibronectin–collagen mix coating (**B**). Type I procollagen expression remains elevated in time on fibronectin coatings and exhibits the lowest expression value on collagen coatings (**C**), which is associated with the increased levels of type I collagen, mostly on fibronectin coatings (**D**). Vitamin C significantly increases the expression levels of col3a1 mRNA over time (**E**), which is associated with the elevated expression of type III collagen on fibronectin–collagen mix (**F**). Vitamin C significantly increases the col5a1 mRNA expression over time, exhibiting a maximum value on fibronectin coatings (**G**). This finding is associated with the increased levels of type V collagen expression on fibronectin coatings, but not on collagen substrates (**H**). Schematic overview of results over time. The solid line towards mature collagen fibers formation serves as a guide to the reader (**I**). Error bars indicate standard error of the mean; * *p* ≤ 0.05; ** *p* ≤ 0.01; *** *p* ≤ 0.001; **** *p* ≤ 0.0001. n = 3 experiments per condition. Fbn (Fibronectin); Coll (Collagen).

**Figure 8 ijms-24-08379-f008:**
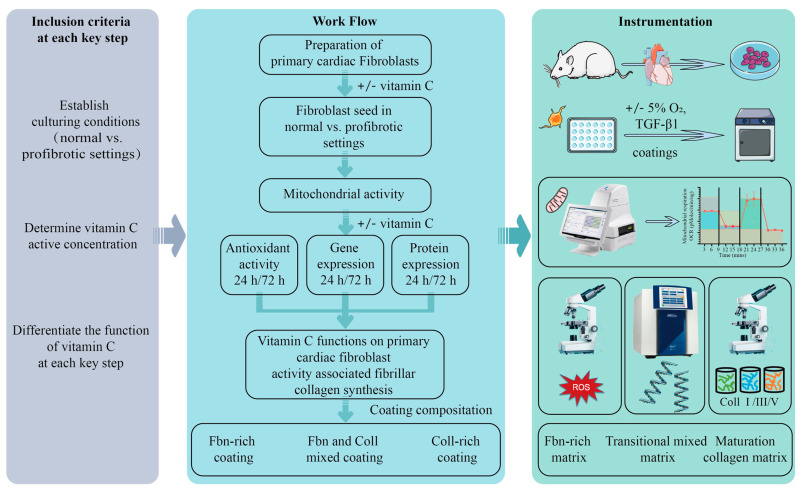
Overview of the workflow, including the inclusion criteria and methodologies.

**Figure 9 ijms-24-08379-f009:**
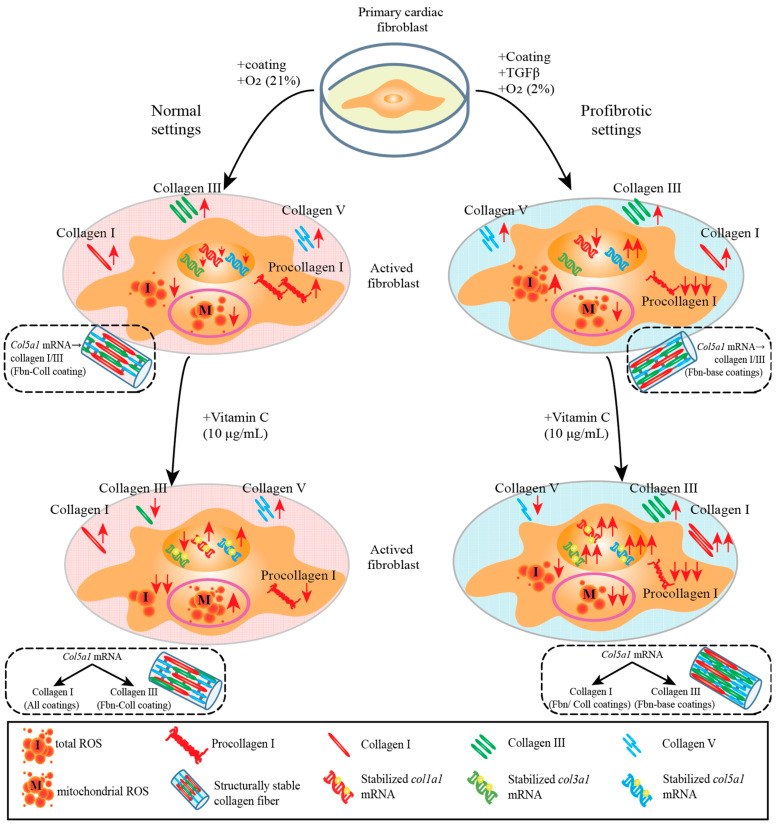
The synergistic and complementary functionalities of vitamin C in in vitro MI replica on fibrillar collagen formation becomes evident at the later incubation time point. In normal conditions, primary cardiac fibroblasts regulate naturally the internal ROS formation to low levels, which are relevant for optimal functionality of cells. The radical scavenger role of vitamin C is highly potentiated over time at the level of the oxidative protection of primary cardiac fibroblasts activation under normal conditions. The synergistic activity of vitamin C and coating composition is clearly visible through the massive upregulation of col5a1 mRNA and downregulation of ROS under profibrotic vs. normal settings. The increase expression of col5a1 mRNA translates into structurally stable type I and III collagen fibers, which is a process that is mainly potentiated by (i) fibronectin–collagen mixture coating in normal settings and (ii) fibronectin and collagen coatings, respectively, in profibrotic conditions. Altogether, optimum supplementation doses of cell media with vitamin C are required to prove its radical scavenging activity and gene and protein stabilizer/modulator role for the synthesis of structurally and compositionally stable collagen fibrils.

**Table 1 ijms-24-08379-t001:** List of primer sequences.

Genes	Primer Sequences	Product Length	Annealing Temp. (°C)	GenBank Numbers
*Col1a1*	5′-GCTCCTCTTAGGGGCCACT (Forward)	103 bp	60 °C	NM_ 007742.4
	5′-CCACGTCTCACCATTGGGG (Reverse)			
*Col3a1*	5′-TCTGAGCTGCTTCTTCCTCTCT (Forward)	98 bp	60 °C	NM_ 009930.2
	5′-GAAGAAACCAGGTTCCACTTTG(Reverse)			
*Col5a1*	5′-GCTACTCCTGTTCCTGCTGC (Forward)	100 bp	60 °C	NM_ 006497644.5
	5′-TGAGGGCAAATTGTGAAAATC (Reverse)			
*β-actin*	5′-AGCCATGTACGTAGCCATCC (Forward)	228 bp	60 °C	NM_ 007393.5
	5′-CTCTCAGCTGTGGTGGTGAA (Reverse)			

## Data Availability

The original contributions presented in the study are included in the article; further inquiries can be directed to the corresponding author.

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
