# Peer review of "Vitamin C Regulates the Profibrotic Activity of Fibroblasts in In Vitro Replica Settings of Myocardial Infarction"

_ijms, 2023, doi:10.3390/ijms24098379_

Round 1

Reviewer 1 Report (Previous Reviewer 1)

I appreciate the authors for responding to my comments. Now the article may be recommended for publication.

Author Response

Please show representative images of DHE- and mitoSOX – DAPI co-staining of primary cardiac fibroblasts.

Answer: The images were added in the Supplementary Figure 2.

Markers of myofibroblast activation should be investigated.

Answer: We have used SM22 as marker for early myofibroblasts activation, as described in the literature. We have now pointed this more clearly in the methods (see Methods 5.4). 

Primers: please add the expected size of the amplicon in bp, the annealing temperatures, GenBank Accession numbers, and verify the absence of non-specific results by blasting the primers. 

Answer: The specificity of the primers was verified with NCBI primer blast tool and analyzed by melting curve to avoid primer dimers and non-specific amplification. These details were now stated and presented in the methods. We have included for the referee the original melting curves (see Methods 5.5, table 1)

Genes

Primer Sequences

Product length

Annealing temp. (°C)

GenBank numbers

Col1a1

5′-GCTCCTCTTAGGGGCCACT (Forward)

103 bp

60℃

NM_

007742.4

5′-CCACGTCTCACCATTGGGG (Reverse)

Col3a1

5′- TCTGAGCTGCTTCTTCCTCTCT (Forward)

98 bp

60℃

NM_

009930.2 

5′-GAAGAAACCAGGTTCCACTTTG(Reverse)

Col5a1

5′- GCTACTCCTGTTCCTGCTGC (Forward)

100 bp

60℃

NM_

006497644.5

5′- TGAGGGCAAATTGTGAAAATC (Reverse)

β-actin

5′- AGCCATGTACGTAGCCATCC (Forward)

228 bp

60℃

NM_

007393.5

5′- CTCTCAGCTGTGGTGGTGAA (Reverse)

Melting curve of COL1A1

Melting curve of COL3A1

Melting curve of COL5A1

Melting curve of β-actin

At least some of the explored pathways should be investigated at the protein level by Western blot.

Answer: We apologies for this inconvenient, however, since the protein material extracted was very low and didn’t allow a proper analysis by western blotting, we have decided to analyze the cells only by immunofluorescence staining. This also presented the advantage to show an eventual change in protein localization or pattern, which was finally not observed. We have now stated this in the method part, to make the results more understandable (see Method 5.4.).

The discussion in its present form fails to interpret the data in the context of what is known in the field: it sounds somehow redundant, as it largely summarizes again data already presented in the Results without placing them in the proper scientific context.

The following pertinent reports should be mentioned/discussed:

Answer: We thank very much this referee for pointed this out. We have now reformulated the discussion to interpret the presented data in the current scientific report, including the suggested references. (see Discussion)

The paper is mainly descriptive and focused on its conclusions, not adequately acknowledging the limitations of the study. The strengths and limitations of the study should be deeply addressed, taking into account sources of potential bias or imprecision: Discuss both direction and magnitude of any potential bias.

Answer: We have now improved the limitation of our study by discussing the data according to any potential bias (see Discussion – Study limitations)

Minor concern:

Please avoid the use of greenish and reddish colors (issues in color blind Readers) in the same figure.

Answer: Thank you for pointing this out. When possible, we have replaced these colors, however, the immunofluorescence images are presented in the original colors.

Reviewer 2 Report (New Reviewer)

Please show representative images of DHE- and mitoSOX – DAPI co-staining of primary cardiac fibroblasts

Markers of myofibroblast activation should be investigated.

Primers: please add the expected size of the amplicon in bp, the annealing temperatures, GenBank Accession numbers, and verify the absence of non-specific results by blasting the primers.

At least some of the explored pathways should be investigated at the protein level by Western blot.

The discussion in its present form fails to interpret the data in the context of what is known in the field: it sounds somehow redundant, as it largely summarizes again data already presented in the Results without placing them in the proper scientific context.

The following pertinent reports should be mentioned/discussed:

PMID: 35563692

PMID: 27340270

PMID: 36140279

PMID: 36498897

PMID: 33287462

PMID: 31981320

PMID: 35868478

PMID: 15254143

PMID: 36145092

PMID: 35007294

PMID: 34905511

The paper is mainly descriptive and focused on its conclusions, not adequately acknowledging the limitations of the study. The strengths and limitations of the study should be deeply addressed, taking into account sources of potential bias or imprecision: Discuss both direction and magnitude of any potential bias.

Minor concern:

Please avoid the use of greenish and reddish colors (issues in color blind Readers) in the same figure.

Author Response

There are a number of key remarks:

  1. It is recommended throughout the text, including in the abstract, to check for transcripts and the use of abbreviations. For example, at the first mention of the words: TGF, ROS, ATP, FCCP, DHE, mitoSOX. It is recommended to shorten the words - myocardial infarction, extracellular matrix.

Answer: Thank you for pointing this out, we have now shortened the words as suggested and carefully scrutinized the paper for any potential abbreviations.

  1. The sentence «While the current stent therapies re-establish the tissue reperfusion and increase the oxygen availability into the injured regions [11], there is no available therapy for managing fibrosis and fibrotic remodeling [12, 13]» is crossed out, in connection with which the sources of literature 11-13 are lost.

Answer: We apologies for the inconvenience, this phrase was deleted and reformulated in the current revised manuscript. The references were carefully verified.

  1. 26 out of 52 references are older than 5 years, it is recommended to revise the list of references. There are a number of recent studies on this topic, for example: doi: 10.12659/MSM.927404, doi: 10.1177/1179546819842804 etc.

Answer: The references were updated in the current form of the revised manuscript.

  1. No parts «C-F» found in figure 1, although they are described in the text.

Answer: We apologies for the misunderstanding, the descripted parts of the figure C-F are from the Supplementary Figure 1. This was now corrected in the revised manuscript.

Non-principled remarks:

  1. In the abstract, it is recommended to correct the phrase «Aim: We aimed to ...» to «Aim: to access ...»

Answer: This was now corrected and replaced.

  1. It is recommended to add «myocardial infarction» to the list of keywords.

Answer: This is now added in the revised manuscript.

  1. When using the phrase, it is recommended to consider replacing «ventricular remodeling» with «left ventricular remodeling» (at the discretion of the authors).

Answer: We have now replaced “ventricular remodeling” with “left ventricular remodeling” as suggested.

  1. It is recommended to remove the word «acute» from the phrase «after acute myocardial infarction» (at the discretion of the authors).

Answer: We agree with this suggestion and we have removed the word “acute” from the description of myocardial infarction.

  1. Figure 9 - fuzzy image, it needs to be edited

Answer: We apologize for this inconvenient, we have now increased the photo resolution and save the image wth 300 ppi (see new Figure 9)

  1. It is recommended after the phrase 4 «Despite the fact that many preclinical studies highlighted the potential of vitamin C to accelerate tissue healing by increasing the synthesis of type I collagen[22], most of the clinical studies target the neutrophils[23], and oxidative stress[24], and have less focused on the effects of vitamin C on the structural remodeling phases of heart such as: inflammation, proliferation, and maturation» remove “2D-scaffolds were designed to replicate the composition of different stages of cardiac extracellular matrix associated with heart tissue remodeling at the inflammatory (fibronectin-rich environment) to-proliferative (fibronectin/collagen mix environment), and maturation phases (collagen-rich environment). We hypothesized that vitamin C may exert in vitro its radical scavenger role and may act as modulator and stabilizer of the intracellular procollagen and extracellular collagen matrix synthesis during healing after myocardial infarction» (authors discretion).

Answer: We thank very much and we agree with this suggestion since deviate the focus of the reader from the message of our study and we have now deleted the suggested text.

  1. It may be worth putting the Materials and Methods section before the results for a better understanding of the information.

Answer: Recent IJMS publications are mainly in IRADM format, which requires the method part to be added at the end of the manuscript.

  1. In figure 1, it is recommended not to duplicate the axis label along the y-axis.

Answer: We have now deleted the axis label, as suggested.

Reviewer 3 Report (New Reviewer)

Article «Vitamin C regulates the profibrotic activity of fibroblasts in in vitro replica settings of myocardial infarction» dedicated to assess systematically vitamin C functions as a potential modulator involved in collagen fibrillogenesis, in an in vitro model mimicking heart tissue healing after acute myocardial infarction.

There are a number of key remarks:

1. It is recommended throughout the text, including in the abstract, to check for transcripts and the use of abbreviations. For example, at the first mention of the words: TGF, ROS, ATP, FCCP, DHE, mitoSOX. It is recommended to shorten the words - myocardial infarction, extracellular matrix.

2. The sentence «While the current stent therapies re-establish the tissue reperfusion and increase the oxygen availability into the injured regions [11], there is no available therapy for managing fibrosis and fibrotic remodeling [12, 13]» is crossed out, in connection with which the sources of literature 11-13 are lost.

3. 26 out of 52 references are older than 5 years, it is recommended to revise the list of references. There are a number of recent studies on this topic, for example: doi: 10.12659/MSM.927404, doi: 10.1177/1179546819842804 etc.

4. No parts «C-F» found in figure 1, although they are described in the text.

Non-principled remarks:

1. In the abstract, it is recommended to correct the phrase «Aim: We aimed to ...» to «Aim: to access ...»

2. It is recommended to add «myocardial infarction» to the list of keywords.

3. When using the phrase, it is recommended to consider replacing «ventricular remodeling» with «left ventricular remodeling» (at the discretion of the authors).

4. It is recommended to remove the word «acute» from the phrase «after acute myocardial infarction» (at the discretion of the authors).

5. Figure 9 - fuzzy image, it needs to be edited

6. It is recommended after the phrase 4 «Despite the fact that many preclinical studies highlighted the potential of vitamin C to accelerate tissue healing by increasing the synthesis of type I collagen[22], most of the clinical studies target the neutrophils[23], and oxidative stress[24], and have less focused on the effects of vitamin C on the structural remodeling phases of heart such as: inflammation, proliferation, and maturation» remove “2D-scaffolds were designed to replicate the composition of different stages of cardiac extracellular matrix associated with heart tissue remodeling at the inflammatory (fibronectin-rich environment) to-proliferative (fibronectin/collagen mix environment), and maturation phases (collagen-rich environment). We hypothesized that vitamin C may exert in vitro its radical scavenger role and may act as modulator and stabilizer of the intracellular procollagen and extracellular collagen matrix synthesis during healing after myocardial infarction» (authors discretion).

7. It may be worth putting the Materials and Methods section before the results for a better understanding of the information.

8. In figure 1, it is recommended not to duplicate the axis label along the y-axis.

Author Response

Please see the attached "Cover Letter"

Round 2

Reviewer 2 Report (New Reviewer)

-

Reviewer 3 Report (New Reviewer)

The authors took into account all the comments and gave exhaustive answers.

Article recommended for publication.

This manuscript is a resubmission of an earlier submission. The following is a list of the peer review reports and author responses from that submission.

Round 1

Reviewer 1 Report

TGF-B dose needs to be corrected in section 2.2. There are numerous typos in  expressing the doses throughout the article.

Quality of the isolated mouse fibroblast needs to be provided in the supplement.

The lowest tested dose of 10 mcg/mL was selected for further investigation as it proved its effeciency at the mitochondrial level both in normoxia and hypoxia. settings. However, it would have been wiser to test further lower doses such as 2, 4, 6, and 8 mcg/mL Vit. C.

Vit. C treatment time: Why 24 and 72 hr treatment conditions were selected?As no statistical comparison were made between the time points (and as shown in fig. 5, there is no difference between them). I feel treatment at two time points are not needed. Also it is very confusing to follow the paper. I recommend the authors to remove (or move to supplement) one of the time points and simplify the graphs in fig 2,3 and 4 for easy understanding.

Figure 2B, y-axis scale needs to be fixed.

Author Response

  1. TGF-B dose needs to be corrected in section 2.2. There are numerous typos in expressing the doses throughout the article.

Response: We thank you for pointing this out. We have now corrected the doses expression of TGF-b1 throughout the entire manuscript.

  1. Quality of the isolated mouse fibroblast needs to be provided in the supplement.

Response: We thank you very much this referee for this observation. Our protocol uses exclusion methods to isolate fibroblasts from other cardiac cell types. Since heart is a highly specialized organ with a limited cell type’s content, cell culture conditions will exclude cardiomyocytes or endothelial cells, as well as blood cells, which otherwise require specific culturing conditions and factors for their differentiation. Therefore, the only cell type from all known cardiac cells which can survive in our culturing conditions are the fibroblasts. For better clarity of the manuscript, and to avoid misunderstandings, now we have added this information in the methods part: “By excluding the cardiac cells, endothelial cells or macrophage/mesenchymal cells, all of which requires special culturing media and factors, as well as by removing the floating cells, we consider the remaining viable cells of fibroblast phenotype in our culture model.” (Method section 5.2).

  1. The lowest tested dose of 10 µg/mL was selected for further investigation as it proved its efficiency at the mitochondrial level both in normoxia and hypoxia. settings. However, it would have been wiser to test further lower doses such as 2, 4, 6, and 8 mcg/mL Vit. C.

Response: We are grateful for pointing this out. We have now discussed the possible biological performance of vitamin C at lower concentrations on ROS-induced cell damage. This paragraph is added now to the section 2.2: “A concentration -function approach of vitamin C excludes smaller concentrations (e.g. 1-2000 mM) at which vitamin C alone showed a poor cell protecting effect (Ref #19). Moreover, at low concentrations the antioxidant function of vitamin C seems to be primarily hindered as it is chemically unstable at the physiological pH (Ref #20).”

  1. C treatment time: Why 24 and 72 hr treatment conditions were selected? As no statistical comparison were made between the time points (and as shown in fig. 5, there is no difference between them). I feel treatment at two time points are not needed. Also it is very confusing to follow the paper. I recommend the authors to remove (or move to supplement) one of the time points and simplify the graphs in fig 2,3 and 4 for easy understanding.

Response: We thank you for these recommendations. We have now introduced in more detail the relevance of 24h and 72h treatment conditions in the methods (see methods section 5.2): “Studies showed that fibroblasts need until 72 hours to change their phenotype and extracellular matrix proteins (Ref #2). Thus, we conducted all analysis in vitro at 24 hours to evaluate potential changes at the gene and mRNA expression, as well as at 72 hours to confirm any modification in protein and extracellular matrix protein expression, but also the downregulation of the genes and mRNA expression, if this is transitory under vitamin C stimulation”. We have also removed the general overview chart (Figure 5) for creating more clearness of the manuscript and reorganized figures 2,3,4 to easier be read and understood (see new Figures 2-7).

  1. Figure 2B, y-axis scale needs to be fixed.

Response: Thank you for pointing this out. The numbers displayed on the OY axis in Figure 2B of this manuscript are of the order of magnitude of 10 lower than the rest values represented in other figures. A modification of maximum value scale in the Figure 2B by a factor of 0.1 would create a different perception to the reader which potentially mislead the interpretation. Therefore, additional information in the Results sections come to clear the discontinuous OY axis in the graph of Fig.2B: “In order to avoid different perceptions, the OY-scale of all graphs in Figure 2 was specified at 0.6” (see Results 2.2, last paragraph)

Reviewer 2 Report

The article „Time-dependent pleiotropic activity of Vitamin C in profibrotic settings mimicking healing after myocardial infarction“ by Yichen Xu et al is a study of possible effects of Vitamin C on different stages of post myocardial infarction healing phase in an exclusively in vitro model. In this model mouse cardiac fibroblasts are seeded on different coatings trying to reflect the environment of the different healing phases. The coating model described by the authors is in general interesting. However, a lot of issues are less than satisfactory, as detailed below:

-        What is the actual hypothesis behind this project? From the introduction it can be assumed that the authors want to provide evidence that vitamin C administration after a heart attack could significantly improve remodelling. This would be an interesting hypothesis, but the present experiments are not very helpful, because the authors could have easily tested this hypothesis in vivo in a myocardial infarction mouse model by vitamin C supplementation. Such in vivo data are completely missing! Instead, mice are only sacrificed to isolate mouse cardiac fibroblasts. If in vitro data alone is relied upon, why are human cardiac fibroblasts not used? This would significantly improve the validity of the study.

-        Essential controls are missing in this project. If the main mechanism of vitamin C is to act as an antioxidant, why are the same experiments not being done with reagents that act as a pro-oxidant and an anti-oxidant? This comparison with other reagents is substantial to draw an adequate conclusion.

-        All experiments have only repetition of n=3. This is too few, especially in critical experiments, to carry out an adequate statistical evaluation. Furthermore, with n=3, the data should be presented in such a way that every data point is visible.

-        The presentation of the data in Figure 2, 3 and 4 is very difficult to follow. Firstly, different quantities are used, sometimes percentages, sometimes scales without any units. Thus, the data cannot be adequately evaluated. Secondly, in the result paragraph, data from normal and profibrotic conditions are compared. Here, a presentation of the data of a target protein next to each other would make much more sense. Moreover, the scales of procollagen I expression are different, which could be misleading.

-        Most of the data presented in the main manuscript is only based on mRNA levels. More protein data is very much needed.

-        The flood of significance asterisks makes it difficult to adequately assess the data in figure 2 to 4. Thus, asterisks sometimes stand alone (what is the comparison group?) or a single group is compared with several groups. According to the methods section, this is always done with the same type of test, a 2-way ANOVA test followed by Newman-Keuls post-hoc comparison. This is hardly comprehensible.

-        For what reason did the authors decide on the time points 24h and 72h?

-        Figure 1: Authors write in the figure description that “at higher concentrations the nonmitochondrial respiration is significantly reduced in normoxia (A) and in hypoxic conditions (B)”. However, figure 1 only shows mitochondrial respiration. The authors do not explain or speculate why higher vitamin C concentrations than 10 µg/ml show a contradictory effect on mitochondrial respiration. However, this is a very important aspect in the clinical context, as it is unclear how much vitamin C would have to be applied to the patient to produce positive effects and not negative consequences. Furthermore, 10 µg/ml is within the range of physiological vitamin C concentration in the blood. Aren't vitamin C administrations in a context of myocardial infarction then perhaps even harmful?

-        Figure 5: What is the rationale behind Figure 5C? 24h and 72h show exactly the same. However, it seems to be one graph with an x and y axis. This is not the way to understand it.

-        For the method part a figure would be helpful so that the reader can understand the method properly.

Minor:

-        Abstract should be divided in paragraphs.

-        Title is unspecific (again, what is the hypothesis?)

-        Page 2, line 4: “[…] dramatically affecting heart´s” – may something is missing here?

-        Fbn and Coll should be explained in the figure description

-        Figure 4, right hand side à Normal instead of Normoal

Author Response

The article „Time-dependent pleiotropic activity of Vitamin C in profibrotic settings mimicking healing after myocardial infarction“ by Yichen Xu et al is a study of possible effects of Vitamin C on different stages of post myocardial infarction healing phase in an exclusively in vitro model. In this model mouse cardiac fibroblasts are seeded on different coatings trying to reflect the environment of the different healing phases. The coating model described by the authors is in general interesting. However, a lot of issues are less than satisfactory, as detailed below:

We gratefully acknowledge the referee’s remarks on our study being interesting and scientifically important and his appreciation on the experimental design. We have now included additional information to address all the remaining issues, significantly improving the quality of our paper, and hoping now to be worth for publication.

-     What is the actual hypothesis behind this project? From the introduction it can be assumed that the authors want to provide evidence that vitamin C administration after a heart attack could significantly improve remodelling. This would be an interesting hypothesis, but the present experiments are not very helpful, because the authors could have easily tested this hypothesis in vivo in a myocardial infarction mouse model by vitamin C supplementation. Such in vivo data are completely missing! Instead, mice are only sacrificed to isolate mouse cardiac fibroblasts. If in vitro data alone is relied upon, why are human cardiac fibroblasts not used? This would significantly improve the validity of the study. 

Response: We apologize for this misunderstanding. We have reformulated now our hypothesis clearer in the introduction and explain the background of the need to use vitamin C for modulating ECM remodeling (see Introduction, paragraph 2 and 3): “Despite the fact that many preclinical studies highlighted the potential of vitamin C to accelerate tissue healing by increasing the synthesis of type I collagen, most of the clinical studies target the neutrophils, and oxidative stress, and have less focus on the effects of vitamin C on the heart fibrosis and extracellular matrix production. Therefore, we plan here to identify the detailed mechanisms and role play of vitamin C in fibrotic processes at the cellular level, in conditions mimicking the healing after myocardial infarction”. We agreed that in vivodata are missing, and we have pointed out this aspect as a limitation of our study. However, since clinical data showed contradictory results using vitamin C in different settings, we intended firstly to study the effect of vitamin C on the fibrosis modulation in the in vitro mimicking settings of healing after myocardial infarction. This is an important step, since the time of administration and the doses could extensively differ at different periods of heart healing and the results of in vivo study could be inconclusive, resulting in unnecessary animal experiments. After performing this study, we have concluded that vitamin C in higher doses can be toxic for myofibroblasts, interfering with the mitochondrial respiration. Lower doses could be benefic not only for ROS clearness directly after myocardial infarction, but also at later time points for extracellular matrix remodeling. These observations are now stated in the discussion and limitation part (see Discussions, last paragraph): “this is a preliminary study for assessing vitamin C role on regulating remodeling in a culture model of healing conditions after myocardial infarction which is aimed to create the context for further in vivo experiments. It seems that the complex pharmacologic analysis to deliver vitamin C in mice to modulate remodeling after infarction requires preliminary in vitro assays to select and predict accurately potential in vivo adverse effects as high doses of vitamin C showed potentially cytotoxic effects on myofibroblasts”. A complex pharmacologic study to deliver vitamin C to mice to modulate remodeling after infarction is in preparation and it will be performed subsequently. In this regard, we have chosen to use mouse fibroblasts in an in vitro myocardial infarction model, to be able to refine and reduce the animal number and to correlate further the in vitro with in vivodata sets for a better clinical translation. Due to the complex and high number of data, we have chosen to publish separately these preliminary in vitro data.

-     Essential controls are missing in this project. If the main mechanism of vitamin C is to act as an antioxidant, why are the same experiments not being done with reagents that act as a pro-oxidant and an anti-oxidant? This comparison with other reagents is substantial to draw an adequate conclusion.

Response: We apologize for this misunderstanding. Our intention is to analyze the role of vitamin C in regulating the fibrotic processes and not oxidative stress, which is broadly known and accepted. We have now changed the introduction (see introduction section, paragraph 2,3: “Despite the fact that many preclinical studies highlighted the potential of vitamin C to accelerate tissue healing by increasing the synthesis of type I collagen, most of the clinical studies target the neutrophils, and oxidative stress, and have less focus on effects of vitamin C on the heart fibrosis and extracellular matrix production. Therefore, we plan here to identify the detailed mechanisms and role play of involved in vitamin C in fibrotic processes at the cellular level in conditions mimicking the healing after myocardial infarction”) and the results part (see results section 2.2.: “These experiments confirmed that at the concentration used in this study, vitamin C keeps its scavenging ROS function”) for clarify this aspect.

-     All experiments have only repetition of n=3. This is too few, especially in critical experiments, to carry out an adequate statistical evaluation. Furthermore, with n=3, the data should be presented in such a way that every data point is visible.

Response: We agree that the sample size is important for the establishment of methodological and significant variations. The first aspect in designing our experiments we considered, was to exclude the influence of one group onto the other study groups in order to exclude/diminish non-meaningful effects on the significance of data points. This would be the case of only 2 replicates. On the other hand, although it is believed that a large number of samples may provide statistical evidence, studies showed that a large number of samples may amplify the detection variance, emphasizing statistical differences which are not relevant.

(Reference #43).

Therefore, we concluded that in the present study the employment of triplicates would provide sufficient significance / data points and it has been introduced in data statistical part (2.7.).

In order to avoid the reduction or amplification detection variance among data, all experiments were performed in triplicate.”

-     The presentation of the data in Figure 2, 3 and 4 is very difficult to follow. Firstly, different quantities are used, sometimes percentages, sometimes scales without any units. Thus, the data cannot be adequately evaluated. Secondly, in the result paragraph, data from normal and profibrotic conditions are compared. Here, a presentation of the data of a target protein next to each other would make much more sense. Moreover, the scales of procollagen I expression are different, which could be misleading.

Response: We have now reorganized the Figures 2,3,4 to make the data more understandable and easier to be read. The new Figures are grouping the same category of data (gene or proteins) and the scale bars or the percentages were uniformized, as recommended (see new Figure 2-7).

-     Most of the data presented in the main manuscript is only based on mRNA levels. More protein data is very much needed. 

Response: We have highlighted now the protein data in separate Figures, to be easier to read and more understandable (see new Figure 2-7).

-     The flood of significance asterisks makes it difficult to adequately assess the data in figure 2 to 4. Thus, asterisks sometimes stand alone (what is the comparison group?) or a single group is compared with several groups. According to the methods section, this is always done with the same type of test, a 2-way ANOVA test followed by Newman-Keuls post-hoc comparison. This is hardly comprehensible.

Response: Thank you for pointing this out, now we have changed the representation in the figures, so the corresponding tests as well as the p values are represented (see new Figures 2-7 and Figure-S1 from supplementary).

-    For what reason did the authors decide on the time points 24h and 72h?

Response: We have now introduced in more detail the relevance of 24h and 72h treatment conditions in the methods (see methods section 5.2): Studies showed that fibroblasts need until 72 hours to change their phenotype and extracellular matrix proteins (Ref #2). Thus, we conducted all analysis in vitro at 24 hours to evaluate potential changes at the gene and mRNA expression, as well as at 72 hours to confirm any modification in protein and extracellular matrix protein expression, but also the downregulation of the genes and mRNA expression, if this is transitory under vitamin C stimulation”.

-     Figure 1: Authors write in the figure description that “at higher concentrations the nonmitochondrial respiration is significantly reduced in normoxia (A) and in hypoxic conditions (B)”. However, figure 1 only shows mitochondrial respiration. The authors do not explain or speculate why higher vitamin C concentrations than 10 µg/ml show a contradictory effect on mitochondrial respiration. However, this is a very important aspect in the clinical context, as it is unclear how much vitamin C would have to be applied to the patient to produce positive effects and not negative consequences. Furthermore, 10 µg/ml is within the range of physiological vitamin C concentration in the blood. Aren't vitamin C administrations in a context of myocardial infarction then perhaps even harmful? 

Response: We try to link hypothetically common characteristics of cells in normoxy and hypoxy conditions including mitochondrial dysfunction at low- and high-doses of vitamin C, which are included in section, Results 2.1. “These findings, might suggest that hypoxia impairs mitochondrial respiration rate by reducing the ATP production. At low doses, vitamin C showed its antioxidant power (radical scavenger) conducting to increased mitochondrial respiration. The cytotoxic effect of vitamin C at high-doses on mitochondrial activity might reside as an effect of changing its functional balance from antioxidant more prone towards co-enzymatic regulator role in collagen synthesis.

We have also clarified the vitamin C concentration findings as a potential preconditioner and combined therapeutic in clinical practice (see Concluding Remarks).  

“Focusing only on single key factor such as antioxidant, it may fail to fully capture the complex relevant role of vitamin C in the clinical practice. Collectively, due to the ample evidence of the temporal complementary functionalities of vitamin C in vitro, the antioxidant-collagen synthetic regulator interplay needs to be further investigated for an optimal dose and timing as a potential preconditioner and combined therapeutic in cardiovascular diseases”.

  • Figure 5: What is the rationale behind Figure 5C? 24h and 72h show the same. However, it seems to be one graph with an x and y axis. This is not the way to understand it.

Response: We agree with this referee that the Figure 5 is very complex and difficult to understand. Since it is not essential for the manuscript and to reduce the amount of information provided by the study, we have decided to remove it from the revised manuscript.

- For the method part a figure would be helpful so that the reader can understand the method properly. 

Response: Many thanks for this recommendation. We have now introduced a new figure representing the methodology for a better understanding of our study (see new Figure 7).

Minor:

- Abstract should be divided in paragraphs.

We have now divided the Abstract in the paragraphs, as suggested.

- Title is unspecific (again, what is the hypothesis?)

We have now changed the title to be more specific for our data and to include a short hypothesis, as suggested: “Vitamin C regulates the profibrotic activity of myofibroblasts in in vitro replica settings of myocardial infarction

- Page 2, line 4: “[…] dramatically affecting heart´s” – may something is missing here?

Thank you for pointing this out. We have now completed the sentence: “…dramatically affecting heart’s biomechanics and function” (see introduction, first paragraph).

- Fbn and Coll should be explained in the figure description

A list with abbreviations was added in each figure legend for a better understanding of the data.

- Figure 4, right hand side à Normal instead of Normoal

Many thanks for pointing out. We have now corrected the writing on the axes. (See revised Figure 2-7)

Round 2

Reviewer 2 Report

Please find additional comments in red below.

The article „Time-dependent pleiotropic activity of Vitamin C in profibrotic settings mimicking healing after myocardial infarction“ by Yichen Xu et al is a study of possible effects of Vitamin C on different stages of post myocardial infarction healing phase in an exclusively in vitro model. In this model mouse cardiac fibroblasts are seeded on different coatings trying to reflect the environment of the different healing phases. The coating model described by the authors is in general interesting. However, a lot of issues are less than satisfactory, as detailed below:

We gratefully acknowledge the referee’s remarks on our study being interesting and scientifically important and his appreciation on the experimental design. We have now included additional information to address all the remaining issues, significantly improving the quality of our paper, and hoping now to be worth for publication.

- What is the actual hypothesis behind this project? From the introduction it can be assumed that the authors want to provide evidence that vitamin C administration after a heart attack could significantly improve remodelling. This would be an interesting hypothesis, but the present experiments are not very helpful, because the authors could have easily tested this hypothesis in vivo in a myocardial infarction mouse model by vitamin C supplementation. Such in vivo data are completely missing! Instead, mice are only sacrificed to isolate mouse cardiac fibroblasts. If in vitro data alone is relied upon, why are human cardiac fibroblasts not used? This would significantly improve the validity of the study.

Response: We apologize for this misunderstanding. We have reformulated now our hypothesis clearer in the introduction and explain the background of the need to use vitamin C for modulating ECM remodeling (see Introduction, paragraph 2 and 3): “Despite the fact that many preclinical studies highlighted the potential of vitamin C to accelerate tissue healing by increasing the synthesis of type I collagen, most of the clinical studies target the neutrophils, and oxidative stress, and have less focus on the effects of vitamin C on the heart fibrosis and extracellular matrix production. Therefore, we plan here to identify the detailed mechanisms and role play of vitamin C in fibrotic processes at the cellular level, in conditions mimicking the healing after myocardial infarction”. We agreed that in vivodata are missing, and we have pointed out this aspect as a limitation of our study. However, since clinical data showed contradictory results using vitamin C in different settings, we intended firstly to study the effect of vitamin C on the fibrosis modulation in the in vitro mimicking settings of healing after myocardial infarction. This is an important step, since the time of administration and the doses could extensively differ at different periods of heart healing and the results of in vivo study could be inconclusive, resulting in unnecessary animal experiments. After performing this study, we have concluded that vitamin C in higher doses can be toxic for myofibroblasts, interfering with the mitochondrial respiration. Lower doses could be benefic not only for ROS clearness directly after myocardial infarction, but also at later time points for extracellular matrix remodeling. These observations are now stated in the discussion and limitation part (see Discussions, last paragraph): “this is a preliminary study for assessing vitamin C role on regulating remodeling in a culture model of healing conditions after myocardial infarction which is aimed to create the context for further in vivo experiments. It seems that the complex pharmacologic analysis to deliver vitamin C in mice to modulate remodeling after infarction requires preliminary in vitro assays to select and predict accurately potential in vivo adverse effects as high doses of vitamin C showed potentially cytotoxic effects on myofibroblasts”. A complex pharmacologic study to deliver vitamin C to mice to modulate remodeling after infarction is in preparation and it will be performed subsequently. In this regard, we have chosen to use mouse fibroblasts in an in vitro myocardial infarction model, to be able to refine and reduce the animal number and to correlate further the in vitro with in vivodata sets for a better clinical translation. Due to the complex and high number of data, we have chosen to publish separately these preliminary in vitro data.

Reviewer:

Unfortunately, the authors' answer on this aspect is not sufficient. The reasoning that the experiments were first carried out with mouse cardiac fibroblasts in order to have to sacrifice fewer mice later is unsatisfactory. The authors already had to sacrifice mice to get the fibroblasts. Thus, these mice could have been fed directly into in vivo experiments. This way, the number of mice increases even more. Furthermore, no justification was provided as to why human fibroblasts were not used instead. At least the most important experiments could be carried out in human fibroblasts to see whether the effect is the same in this cell type. Instead, the authors have now already moved on to an in vivo mouse model based on these data. Many translational studies have failed precisely because of this aspect, that results from the animal model could not ultimately be transferred to the human organism. Why then not use this possibility of human cardiac fibroblasts? Performing such experiments with human cardiac fibroblasts would improve the study significantly.

- Essential controls are missing in this project. If the main mechanism of vitamin C is to act as an antioxidant, why are the same experiments not being done with reagents that act as a pro-oxidant and an anti-oxidant? This comparison with other reagents is substantial to draw an adequate conclusion.

Response: We apologize for this misunderstanding. Our intention is to analyze the role of vitamin C in regulating the fibrotic processes and not oxidative stress, which is broadly known and accepted. We have now changed the introduction (see introduction section, paragraph 2,3: “Despite the fact that many preclinical studies highlighted the potential of vitamin C to accelerate tissue healing by increasing the synthesis of type I collagen, most of the clinical studies target the neutrophils, and oxidative stress, and have less focus on effects of vitamin C on the heart fibrosis and extracellular matrix production. Therefore, we plan here to identify the detailed mechanisms and role play of involved in vitamin C in fibrotic processes at the cellular level in conditions mimicking the healing after myocardial infarction”) and the results part (see results section 2.2.: “These experiments confirmed that at the concentration used in this study, vitamin C keeps its scavenging ROS function”) for clarify this aspect.

- All experiments have only repetition of n=3. This is too few, especially in critical experiments, to carry out an adequate statistical evaluation. Furthermore, with n=3, the data should be presented in such a way that every data point is visible.

Response: We agree that the sample size is important for the establishment of methodological and significant variations. The first aspect in designing our experiments we considered, was to exclude the influence of one group onto the other study groups in order to exclude/diminish non-meaningful effects on the significance of data points. This would be the case of only 2 replicates. On the other hand, although it is believed that a large number of samples may provide statistical evidence, studies showed that a large number of samples may amplify the detection variance, emphasizing statistical differences which are not relevant.

(Reference #43).

Therefore, we concluded that in the present study the employment of triplicates would provide sufficient significance / data points and it has been introduced in data statistical part (2.7.).

In order to avoid the reduction or amplification detection variance among data, all experiments were performed in triplicate.”

Reviewer:

This is a very weak argumentation method, as a high standard deviation can already be seen in some samples. Moreover, applying these hypothesis tests to such a small number of samples is of limited power. Therefore, an increase in the n numbers would have been desirable, especially in the trials with very high standard deviation, but unfortunately this was not implemented.

Now the authors have included data points in the graphs, which was generally very necessary. Now it seems very incomprehensible that in some graphs a different number of data points appear, when the authors stated that each experiment was conducted n=3 in triplicates.

The range here is from 2 data points (e.g. Control 3H) to 10 data points (e.g. Coll 3D). How is this possible?

- The presentation of the data in Figure 2, 3 and 4 is very difficult to follow. Firstly, different quantities are used, sometimes percentages, sometimes scales without any units. Thus, the data cannot be adequately evaluated. Secondly, in the result paragraph, data from normal and profibrotic conditions are compared. Here, a presentation of the data of a target protein next to each other would make much more sense. Moreover, the scales of procollagen I expression are different, which could be misleading.

Response: We have now reorganized the Figures 2,3,4 to make the data more understandable and easier to be read. The new Figures are grouping the same category of data (gene or proteins) and the scale bars or the percentages were uniformized, as recommended (see new Figure 2-7).

- Most of the data presented in the main manuscript is only based on mRNA levels. More protein data is very much needed.

Response: We have highlighted now the protein data in separate Figures, to be easier to read and more understandable (see new Figure 2-7).

- The flood of significance asterisks makes it difficult to adequately assess the data in figure 2 to 4. Thus, asterisks sometimes stand alone (what is the comparison group?) or a single group is compared with several groups. According to the methods section, this is always done with the same type of test, a 2-way ANOVA test followed by Newman-Keuls post-hoc comparison. This is hardly comprehensible.

Response: Thank you for pointing this out, now we have changed the representation in the figures, so the corresponding tests as well as the p values are represented (see new Figures 2-7 and Figure-S1 from supplementary).

- For what reason did the authors decide on the time points 24h and 72h?

Response: We have now introduced in more detail the relevance of 24h and 72h treatment conditions in the methods (see methods section 5.2): Studies showed that fibroblasts need until 72 hours to change their phenotype and extracellular matrix proteins (Ref #2). Thus, we conducted all analysis in vitro at 24 hours to evaluate potential changes at the gene and mRNA expression, as well as at 72 hours to confirm any modification in protein and extracellular matrix protein expression, but also the downregulation of the genes and mRNA expression, if this is transitory under vitamin C stimulation”.

- Figure 1: Authors write in the figure description that “at higher concentrations the nonmitochondrial respiration is significantly reduced in normoxia (A) and in hypoxic conditions (B)”. However, figure 1 only shows mitochondrial respiration. The authors do not explain or speculate why higher vitamin C concentrations than 10 µg/ml show a contradictory effect on mitochondrial respiration. However, this is a very important aspect in the clinical context, as it is unclear how much vitamin C would have to be applied to the patient to produce positive effects and not negative consequences. Furthermore, 10 µg/ml is within the range of physiological vitamin C concentration in the blood. Aren't vitamin C administrations in a context of myocardial infarction then perhaps even harmful?

Response: We try to link hypothetically common characteristics of cells in normoxy and hypoxy conditions including mitochondrial dysfunction at low- and high-doses of vitamin C, which are included in section, Results 2.1. “These findings, might suggest that hypoxia impairs mitochondrial respiration rate by reducing the ATP production. At low doses, vitamin C showed its antioxidant power (radical scavenger) conducting to increased mitochondrial respiration. The cytotoxic effect of vitamin C at high-doses on mitochondrial activity might reside as an effect of changing its functional balance from antioxidant more prone towards co-enzymatic regulator role in collagen synthesis”.

We have also clarified the vitamin C concentration findings as a potential preconditioner and combined therapeutic in clinical practice (see Concluding Remarks).

“Focusing only on single key factor such as antioxidant, it may fail to fully capture the complex relevant role of vitamin C in the clinical practice. Collectively, due to the ample evidence of the temporal complementary functionalities of vitamin C in vitro, the antioxidant-collagen synthetic regulator interplay needs to be further investigated for an optimal dose and timing as a potential preconditioner and combined therapeutic in cardiovascular diseases”.

  • Figure 5: What is the rationale behind Figure 5C? 24h and 72h show the same. However, it seems to be one graph with an x and y axis. This is not the way to understand it.

Response: We agree with this referee that the Figure 5 is very complex and difficult to understand. Since it is not essential for the manuscript and to reduce the amount of information provided by the study, we have decided to remove it from the revised manuscript.

Reviewer:

Just removing the figure without providing a better figure is not the best option. A graphic that pictorially summarises the most important results of the study is necessary.

- For the method part a figure would be helpful so that the reader can understand the method properly.

Response: Many thanks for this recommendation. We have now introduced a new figure representing the methodology for a better understanding of our study (see new Figure 7).

Reviewer:

There is no Figure 7 in the manuscript.

Minor:

- Abstract should be divided in paragraphs.

We have now divided the Abstract in the paragraphs, as suggested.

- Title is unspecific (again, what is the hypothesis?)

We have now changed the title to be more specific for our data and to include a short hypothesis, as suggested: “Vitamin C regulates the profibrotic activity of myofibroblasts in in vitro replica settings of myocardial infarction

- Page 2, line 4: “[…] dramatically affecting heart´s” – may something is missing here?

Thank you for pointing this out. We have now completed the sentence: “…dramatically affecting heart’s biomechanics and function” (see introduction, first paragraph).

- Fbn and Coll should be explained in the figure description

A list with abbreviations was added in each figure legend for a better understanding of the data.

- Figure 4, right hand side à Normal instead of Normoal

Many thanks for pointing out. We have now corrected the writing on the axes. (See revised Figure 2-7)

Author Response

  1. Unfortunately, the authors' answer on this aspect is not sufficient. The reasoning that the experiments were first carried out with mouse cardiac fibroblasts in order to have to sacrifice fewer mice later is unsatisfactory. The authors already had to sacrifice mice to get the fibroblasts. Thus, these mice could have been fed directly into in vivo experiments. This way, the number of mice increases even more. Furthermore, no justification was provided as to why human fibroblasts were not used instead. At least the most important experiments could be carried out in human fibroblasts to see whether the effect is the same in this cell type. Instead, the authors have now already moved on to an in vivo mouse model based on these data. Many translational studies have failed precisely because of this aspect, that results from the animal model could not ultimately be transferred to the human organism. Why then not use this possibility of human cardiac fibroblasts? Performing such experiments with human cardiac fibroblasts would improve the study significantly.

Response: We appreciate the referee’s remarks. The fibroblasts used in this study were isolated from heart of mice which were used to be donors for all kind of cells types, for all research groups in the institute. The number of these mice were very reduced, since the proliferation rate of the isolated cardiac fibroblasts is very high. However, we have preferred to use cardiac fibroblasts and not just normal fibroblasts, as will be available for humans, since there are many recent studies indicating the significant differences between cardiac fibroblasts and other types of fibroblasts. This has been indicated in the section 5.2.: Moreover, we have preferred to use cardiac fibroblasts and not just normal fibroblasts, as will be available for humans, since there are many recent studies indicating the significant differences between cardiac fibroblasts and other types of fibroblasts. (References:#42-#44)“Moreover, currently, the PI has relocated in other country and additional animal experiments cannot be performed at this time point. We apologize for this inconvenience, and we leave to the reviewers and editors’ opinions the decision of manuscript publication under these conditions.

  1. This is a very weak argumentation method, as a high standard deviation can already be seen in some samples. Moreover, applying these hypothesis tests to such a small number of samples is of limited power. Therefore, an increase in the n numbers would have been desirable, especially in the trials with very high standard deviation, but unfortunately this was not implemented. Now the authors have included data points in the graphs, which was generally very necessary. Now it seems very incomprehensible that in some graphs a different number of data points appear, when the authors stated that each experiment was conducted n=3 in triplicates. The range here is from 2 data points (e.g. Control 3H) to 10 data points (e.g. Coll 3D). How is this possible?

Response: We apologize for this misunderstanding, and we have accordingly corrected it in the materials & method section (section 5). We have now introduced all relevant data points for a clear understanding (see Figure 3H). “Data points of immunofluorescent images illustrating reduced fibroblasts activity are not considered for further analysis. As such the reduction or amplification detection variance among all data will be avoided”.

  1. Just removing the figure without providing a better figure is not the best option. A graphic that pictorially summarises the most important results of the study is necessary.

Response: We thanks for this recommendation. Now we have summarized the relevant results in the new Figure 7.

  1. There is no Figure 7 in the manuscript.

Response: The old figure 7 represents a summary of the methodology and workflow for a better understanding of our study (section 5). The old Figure 7 becomes now Figure 8.